# Head-Level Mechanistic Attribution for Hallucination Control: Training-Free Counteractive Pruning in LVLMs

## Abstract

Large vision-language models (LVLMs) excel at multimodal tasks but often generate instance-level object hallucinations, describing nonexistent objects. Since existing methods overlook functional conflicts within attention heads and lack principled, fine-grained attribution and intervention at the head level, hallucination suppression is often accompanied by a substantial loss of semantic informativeness. To overcome these limitations, we propose HACP, a unified framework that enables fine-grained internal hallucination control via precise intervention at the attention head level. Specifically, we introduce InfoSpectralScore, a novel attribution metric based on eigendecomposition with spectral variance and entropy penalties, which allows for the accurate identification of hallucination-inducing heads. We further develop a dynamic, training-free pruning strategy that adaptively suppresses hallucination-prone heads while reinforcing faithful heads during inference. Extensive experiments across multiple LVLMs and benchmarks demonstrate that HACP achieves state-of-the-art hallucination mitigation, substantially reducing hallucinations while better preserving caption informativeness compared to existing approaches, thus offering a robust and transferable solution for controllable and interpretable multimodal generation. The source code will be released upon acceptance.

## 1 Introduction

Large vision–language models (LVLMs) Yin et al. (2024a); Liu et al. (2024c;a); Chen et al. (2023) are driving significant progress in image captioning, visual question answering, and multimodal dialogue. However, these models often generate *instance-level object hallucinations*, describing objects that are not present in the input image. Such errors critically undermine their reliability in safety-sensitive applications.

While numerous hallucination mitigation techniques have been proposed, such as instruction tuning, contrastive decoding, retrieval augmentation, and enhanced visual grounding—these methods largely operate at the output level. Consequently, existing approaches often struggle with the longstanding trade-off between hallucination suppression and semantic informativeness Liu et al. (2024d); He et al. (2025).

Recent advances in mechanistic interpretability have revealed that large language models' behaviors can be traced to specific internal circuits and representational components. In LVLMs, multi-head attention enables cross-modal fusion and semantic abstraction, but most existing approaches treat attention monolithically or only at the layer level. This overlooks a crucial fact: individual attention heads may encode distinct semantic functions, and growing evidence shows that some heads disproportionately contribute to faithful or hallucinatory outputs Bi et al. (2024); Geng et al. (2023); Jiang et al. (2025); Chen et al. (2024a). Such functional conflicts among attention heads may underpin the longstanding trade-off between hallucination suppression and semantic informativeness, yet current approaches rarely provide a principled, fine-grained attribution and intervention framework at the head level to resolve this challenge Liu et al. (2024d); He et al. (2025).

To fill this gap, we propose **HACP**, a unified head-level dynamic pruning framework for LVLMs, drawing on insights from mechanistic interpretability. Central to our approach are two key innovations: (1) *InfoSpectralScore*, a semantics-based attribution metric that enables systematic identification of faithful and hallucination-prone attention heads; and (2) a dynamic, training-free counteractive pruning strategy that adaptively suppresses hallucination-prone heads while reinforcing faithful ones during inference.

As illustrated in Figure 1, our framework enables automated, fine-grained internal control, resulting in more accurate and faithful captions. Extensive experiments across diverse LVLMs and benchmarks demonstrate that our method achieves state-of-the-art performance, substantially reducing hallucinations and better preserving caption informativeness compared to existing approaches.

Our main contributions are as follows:

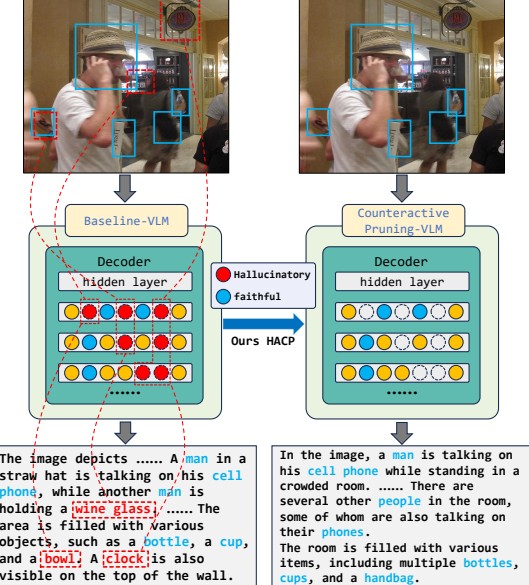

**Figure 1:** Object Hallucination Mitigation via Head-Level Pruning. HACP reduces hallucinations by pruning attention heads associated with spurious outputs, leading to more faithful captions than the baseline LVLM.

- We present a novel **head-level** hallucination mitigation framework for LVLMs. Our method leverages insights from interpretability to enable fine-grained, instance-specific control at inference time, without requiring any retraining.
- We propose *InfoSpectralScore*, a semantics-based attribution metric for attention heads, enabling systematic identification of both hallucination-prone and faithful heads.
- We develop a dynamic, training-free pruning strategy that adaptively suppresses hallucination-inducing heads while reinforcing faithful ones during inference.
- Extensive experiments show that our approach not only reduces hallucination rates and preserves semantic expressivity, but also generalizes robustly across distributions and tasks.

## 2 RELATED WORK

LVLMs are designed to generate semantically grounded descriptions of visual inputs. Yet recent studies consistently show that they often describe nonexistent objects, a phenomenon known as object hallucination Rohrbach et al. (2019); Liu et al. (2024b). Such errors occur at the instance (spurious objects), attribute (incorrect properties), and spatial (misplaced relations) levels, undermining reliability in high-stakes domains such as assistive technology and autonomous systems.

Previous attempts to mitigate hallucinations largely relied on output-level or decoding-oriented adjustments. Instruction tuning with curated image–text pairs Zhang et al. (2024); Liu et al. (2024a); Dai et al. (2023); Li et al. (2023a) and grounding-aware objectives Favero et al. (2024) improved general visual reasoning but offered limited control over hallucinations Geigle et al. (2024). Contrastive, grounding-aware, and guided-decoding strategies further constrained generation using external visual signals or exemplar-based guidance Deng et al. (2024); Leng et al. (2023); Qu et al. (2024); Chen et al. (2024b); Xie et al. (2024); Li et al. (2025); Zhao et al. (2025). These approaches reduce hallucinations during inference but often compromise output diversity or informativeness. External expert-based systems take a different path: instead of steering the decoding process, they introduce additional correction networks or logic-aware verifiers to post-edit model outputs Yin et al. (2024b); Wu et al. (2024); Yu et al. (2024). These expert modules often improve reliability but at the cost of increased architectural complexity and inference latency.

Beyond such output-level techniques, advances in mechanistic interpretability enabled the decomposition of deep models into interpretable circuits and, together with activation patching, localized causal associations in language models Nanda et al. (2023); Elhage et al. (2022); Wang et al. (2022); García-Carrasco et al. (2024); Azaria & Mitchell (2023); Meng et al. (2023). For vision–language models, attention-based attribution and cross-modal analyses became prominent Bi et al. (2024); Geng et al. (2023); Gandelsman et al. (2024), yet most strategies relied on simple heuristics or layer-level aggregation Sarkar et al. (2025); He et al. (2025), lacking a semantics-grounded head-level attribution framework. Recent works further revealed that specific attention heads in intermediate layers are particularly responsible for hallucinations Jiang et al. (2025); Chen et al. (2024a); Yang et al. (2025b), motivating the development of more fine-grained and interpretable intervention mechanisms.

## 3 Method

### 3.1 Motivation

Despite remarkable progress, LVLMs still suffer from instance-level hallucinations, which recent evidence Yang et al. (2025a); He et al. (2025); Sarkar et al. (2025) links to functional conflicts among attention heads: some promote faithful grounding, while others introduce spurious objects. These conflicts appear central to the longstanding trade-off between hallucination suppression and semantic informativeness. Yet current methods lack principled, fine-grained mechanisms for diagnosing and intervening at the head level.

First, existing approaches rarely provide semantics-grounded metrics to attribute hallucinations to specific attention heads. Output-level measures, such as Kullback–Leibler (KL) divergence,

$$D_{\mathrm{KL}}(p \parallel q) = \sum_i p_i \log\left(\frac{p_i}{q_i}\right) \tag{1}$$

only capture distributional shifts after ablation but cannot reveal whether the change improves semantic alignment. Both faithful and hallucination-prone heads may yield high KL scores, limiting diagnostic utility.

Inspired by research in LLMs Chen et al. (2024a) and empirical analyses of hallucination-associated heads in LVLMs Jiang et al. (2025), we propose *InfoSpectralScore*: a semantics-based, head-level attribution metric that systematically quantifies the semantic capacity of each attention head in targeted intermediate layers, enabling precise identification of both hallucination-prone and faithful heads.

Second, conventional head-pruning strategies (e.g., zero or mean ablation) are often too severe, over-penalizing the model and degrading semantic expressivity. To address this, we develop Training-Free Counteractive Pruning: a dynamic, inference-time intervention that suppresses hallucination-inducing heads while reinforcing faithful ones, achieving hallucination mitigation without compromising output quality or semantic richness.

After addressing these challenges, we introduce a unified pipeline for head-level attribution and inference-time hallucination mitigation (Figure 2). Building on this foundation, we further design an automated, instance-adaptive mechanism, with implementation details given in Section 3.4.

### 3.2 Semantic Attribution via InfoSpectralScore

To isolate the causal effect of each attention head, we employ Sequential Head Pruning (SHPM), traversing all heads in target layers $\mathcal{L}$(see Algorithm 1). For each head, we attach a mean-replacement hook, run inference on an attribution set $\mathcal{D}$, and collect final-token embeddings. Stacking these for $K$ samples yields $Z \in \mathbb{R}^{K \times d}$. The covariance matrix is

$$\Sigma = \frac{1}{K-1}(Z - \bar{Z})^\top (Z - \bar{Z}) + \epsilon I, \tag{2}$$

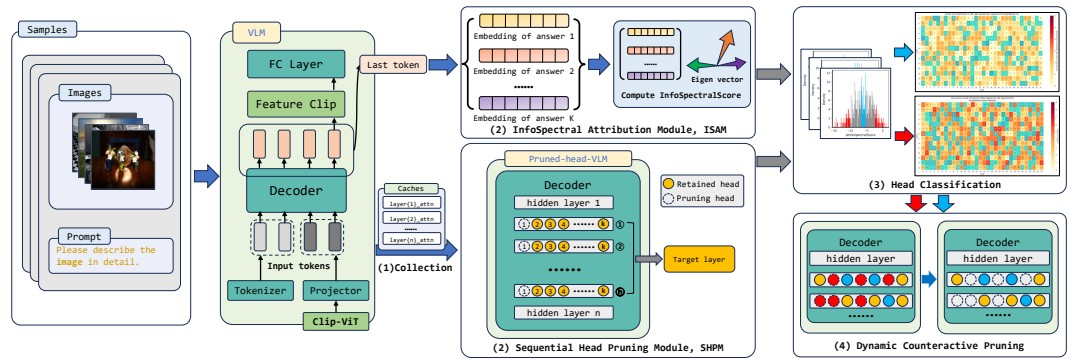

**Figure 2: Unified pipeline for head-level attribution and inference-time hallucination mitigation.** Our framework consists of four stages: (1) activation patching and cache collection, (2) head-level attribution via InfoSpectralScore, (3) faithful vs. hallucinatory head classification, and (4) dynamic counteractive pruning during inference.

---

**Algorithm 1** Head-Level Attribution via SHPM and ISAM

---

**Require:** Model $M$, attribution set $\mathcal{D}$, target layers $\mathcal{L}$, per-layer head counts $\{n_h(l)\}_{l \in \mathcal{L}}$, rounds $R$

**Ensure:** Attribution scores $\mathcal{S}_{l,j}$; per-layer faithful/hallucinatory head sets $\mathcal{H}_{\text{faith}}^{(l)}, \mathcal{H}_{\text{hallu}}^{(l)}$

1: **for** round $r = 1$ to $R$ **do**
2:    **for** each layer $l \in \mathcal{L}$ **do**
3:       **for** head index $j = 1$ to $n_h(l)$ **do**
4:          Prune head $(l, j)$, run inference on $\mathcal{D}$
5:          Compute InfoSpectralScore$(l, j)$
6:          Append to $\mathcal{S}_{l,j}$
7:       **end for**
8:    **end for**
9: **end for**
10: Aggregate $\mathcal{S}_{l,j}$ across rounds
11: Classify heads by thresholds to obtain $\mathcal{H}_{\text{faith}}^{(l)}, \mathcal{H}_{\text{hallu}}^{(l)}$
12: **return** $\{\mathcal{S}_{l,j}\}, \{\mathcal{H}_{\text{faith}}^{(l)}\}, \{\mathcal{H}_{\text{hallu}}^{(l)}\}$

---

with eigenvalues $\lambda_1 \geq \cdots \geq \lambda_d$. The *EigenScore* is defined as

$$\text{EigenScore} = \sum_{i=1}^{k^*} \lambda_i, \tag{3}$$

where $k^*$ is the smallest integer such that $\sum_{i=1}^{k^*} \lambda_i \geq \tau \sum_{j=1}^{d} \lambda_j$ (e.g., $\tau = 0.9$).

However, EigenScore alone may miss pathologies such as *anisotropy* and *sparsity* in hallucination-prone heads. We thus introduce the InfoSpectral Attribution Module (ISAM), which augments EigenScore with two regularizers:

$$\text{SpectralVar} = \text{Var}\big(\log \lambda_1, \ldots, \log \lambda_d\big) \tag{4}$$

which penalizes anisotropy by measuring dispersion in the log-spectrum, where $d$ is the total number of eigenvalues.

$$p_i = \frac{\lambda_i}{\sum_{j=1}^{d} \lambda_j} \tag{5}$$

$$\text{SpectralEntropy} = -\sum_{i=1}^{d} p_i \log\big(p_i + \epsilon\big) \tag{6}$$

---

**Algorithm 2** Dynamic Counterbalance of Attention Heads

---

**Require:** For each layer: faithful set $\mathcal{H}_{\text{faith}}$, hallucinatory set $\mathcal{H}_{\text{hallu}}$, head outputs $\{\mathbf{h}_j\}$; modulation $\mu$, smoothing $\lambda$

1: Remove overlaps: $\mathcal{H}'_{\text{hallu}} \leftarrow \mathcal{H}_{\text{hallu}} \setminus \mathcal{H}_{\text{faith}}$
2: **for** each layer **do**
3:   **if** $|\mathcal{H}_{\text{faith}}| < 2$ or $|\mathcal{H}'_{\text{hallu}}| < 2$ **then**
4:     **continue**
5:   **end if**
6:   $\mathbf{v}_f \leftarrow \frac{1}{|\mathcal{H}_{\text{faith}}|} \sum_{j \in \mathcal{H}_{\text{faith}}} \mathbf{h}_j, \quad \mathbf{v}_h \leftarrow \frac{1}{|\mathcal{H}'_{\text{hallu}}|} \sum_{j \in \mathcal{H}'_{\text{hallu}}} \mathbf{h}_j$
7:   $D \leftarrow \|\mathbf{v}_h\|/(\|\mathbf{v}_f\| + \epsilon), \quad \tilde{D} \leftarrow \lambda \tilde{D}_{\text{prev}} + (1 - \lambda)D$
8:   **for** each head $j$ **do**
9:     **if** $j \in \mathcal{H}_{\text{faith}}$ **then**
10:       $\mathbf{h}_j \leftarrow \mathbf{h}_j + \mu \tilde{D} \mathbf{v}_f$
11:     **else if** $j \in \mathcal{H}'_{\text{hallu}}$ **then**
12:       $\mathbf{h}_j \leftarrow \mathbf{h}_j - \mu \tilde{D} \mathbf{v}_h$
13:     **end if**
14:   **end for**
15: **end for**

---

which quantifies sparsity by the entropy of the normalized spectrum, where $\epsilon$ is a small constant (e.g., $10^{-8}$) for numerical stability.

Combining these, the InfoSpectralScore is defined as:

$$\text{InfoSpectralScore} = \text{EigenScore} - \alpha \,\text{SpectralVar} - \gamma \,\text{SpectralEntropy}, \tag{7}$$

where $\alpha$ and $\gamma$ are regularization weights. Hyperparameters are selected via grid search (see Appendix B).

### 3.3 COUNTERBALANCING ATTENTION HEADS VIA DIRECTIONAL PRUNING

To suppress hallucination-inducing behaviors while preserving faithful generation, we introduce a dynamic counteractive pruning mechanism (see Algorithm 2). Let $\mathcal{H}_{\text{faith}}$ and $\mathcal{H}_{\text{hallu}}$ denote the sets of faithful and hallucinatory heads identified by ISAM. We first remove overlaps to avoid ambiguous signals:

$$\mathcal{H}'_{\text{hallu}} = \mathcal{H}_{\text{hallu}} \setminus \mathcal{H}_{\text{faith}}. \tag{8}$$

For each layer, let $\mathbf{h}_j$ be the output vector of head $j$ at the final token position. We compute the per-set mean directions

$$\mathbf{v}_f = \frac{1}{|\mathcal{H}_{\text{faith}}|} \sum_{j \in \mathcal{H}_{\text{faith}}} \mathbf{h}_j, \qquad \mathbf{v}_h = \frac{1}{|\mathcal{H}'_{\text{hallu}}|} \sum_{j \in \mathcal{H}'_{\text{hallu}}} \mathbf{h}_j. \tag{9}$$

The modulation ratio is

$$D = \frac{\|\mathbf{v}_h\|}{\|\mathbf{v}_f\| + \epsilon}, \tag{10}$$

and its exponentially smoothed version is

$$\tilde{D} = \lambda \tilde{D}_{\text{prev}} + (1 - \lambda) D. \tag{11}$$

Finally, each head $j$ is reweighted as

$$\mathbf{h}'_j = \begin{cases} \mathbf{h}_j + \mu \tilde{D} \mathbf{v}_f, & j \in \mathcal{H}_{\text{faith}}, \\ \mathbf{h}_j - \mu \tilde{D} \mathbf{v}_h, & j \in \mathcal{H}'_{\text{hallu}}, \\ \mathbf{h}_j, & \text{otherwise.} \end{cases} \tag{12}$$

We skip any layer with fewer than two heads in either set. We implement the mechanism via forward hooks at inference, reshaping the post-attention tensor to $[B, T, H, D]$, computing $\mathbf{v}_f, \mathbf{v}_h$, updating $\tilde{D}$, and applying Eq. (12) on GPU without extra parameters. Hyperparameters $\mu$ and $\lambda$ are selected via grid search (Appendix C).

---

**Algorithm 3** Task-Specific Pruning Pipeline for a Single Image–Prompt Pair

---

**Require:** LVLM $M$, image $I$, prompt $P$, BO budget $T$
**Ensure:** Pruned caption $\hat{y}_p$
 1: **Attribution (Algorithm 1):** run head-level ablations on $(I, P)$ to obtain InfoSpectralScore for all heads and classify faithful/hallucinatory sets.
 2: Run baseline inference to obtain $\hat{y}_0$ and metrics $m_0$.
 3: **for** $t = 1$ to $T$ **do**
 4:   Use Bayesian optimization to propose pruning parameters $(\mu_t, \lambda_t)$.
 5:   Instantiate the counterbalance module (Algorithm 2) using $(\mu_t, \lambda_t)$ and the attribution-derived head sets.
 6:   Run $M$ with this configuration to obtain $\hat{y}_t$ and metrics $m_t$.
 7:   Update BO surrogate using hallucination-aware objective $J(m_t, m_0)$.
 8: **end for**
 9: Let $(\mu^\star, \lambda^\star)$ be the best BO parameters.
10: Apply Algorithm 2 with $(\mu^\star, \lambda^\star)$ to produce the final caption $\hat{y}_p$.

---

### 3.4 Unified Automation and Task-Specific Adaptation Pipeline

In practical LVLM deployments, hallucination behavior varies across inputs, so each image–prompt pair requires on-the-fly adaptation. To support thisinstance-level behavior, we integrate the attribution module (Section 3.2) with the counterbalance mechanism (Section 3.3) into a unified, task-specific pruning pipeline. Algorithm 3 outlines the procedure. For a given input, the pipeline first performs head-level attribution to identify faithful and hallucinatory directions. It then runs a small Bayesian optimization loop to search over the pruning parameters $(\mu, \lambda)$ under a hallucination-aware objective, and applies the best configuration to generate the final caption. This design enables each instance to receive a lightweight, attribution-guided adjustment tailored to its own hallucination profile. Appendix A.2 details the two engineering improvements that ensure the pipeline remains efficient and robust in practice.

## 4 Experiment

### 4.1 Experimental Setup

We evaluate our approach on three representative LVLMs: LLaVA-1.5 (7B, 13B), Shikra-7B, and Qwen2.5-VL-7B-Instruct Wang et al. (2024). Results in Sections 4.2 and 4.3 are based on the generalization pipeline, whereas all comparisons against other methods (reported as "Ours") use the full task-specific pruning pipeline described in Algorithm 3. We additionally compare against competitive training-free hallucination-mitigation methods, including VCD Leng et al. (2023), OPERA Huang et al. (2024), PAI Liu et al. (2024d), SPIN Sarkar et al. (2025), MLIH Jiang et al. (2025), and VHR He et al. (2025), each evaluated using its officially recommended settings.

For evaluation, we follow the CHAIR protocol Rohrbach et al. (2019) on the COCO 2014 dataset Lin et al. (2015), reporting $\text{CHAIR}_s$, $\text{CHAIR}_i$, recall, precision, F1, and caption length. On POPE Li et al. (2023b), we report accuracy, precision, recall, and F1. To assess broader multimodal capabilities beyond object hallucination, we further evaluate on MME Fu et al. (2025) and HallusionBench Guan et al. (2024), applying our method only to LLaVA-1.5 for these two benchmarks.

All configurations and implementation details for both our method and competing baselines are provided in Appendix A.1. The appendix also includes a report of the computational cost for every method.

### 4.2 Generalization Pipeline: Systematic Cross-Model Evaluation

The generalization pipeline evaluates whether head-level attribution and pruning remain effective across different model architectures and data distributions. Unlike the task-specific

**Table 1:** Relative improvement (%) on hallucination metrics. Each cell is Delta (%) from Baseline to Pruned. **Bold numbers** indicate the best performance for each metric within each model and evaluation set block.

| Model | Group | Validation Δ (%) | | | Hallucination Δ (%) | | | Generalization Δ (%) | | |
|---|---|---|---|---|---|---|---|---|---|---|
| | | CHAIR$_s$ | CHAIR$_i$ | F1 | CHAIR$_s$ | CHAIR$_i$ | F1 | CHAIR$_s$ | CHAIR$_i$ | F1 |
| LLaVA-7B | [5,18] | -6.85 | -1.96 | +0.60 | **-11.32** | -3.81 | **+1.31** | -4.11 | **-5.72** | +0.37 |
| | [19,26] | **-8.58** | **-7.27** | **+1.35** | -5.24 | -3.16 | +0.73 | -3.61 | -5.02 | **+1.63** |
| | Merged | -8.05 | -3.29 | -0.04 | -9.77 | **-4.34** | -0.07 | **-6.25** | -5.55 | +0.55 |
| LLaVA-13B | [5,18] | -7.41 | **-12.03** | +2.27 | -7.89 | -2.57 | -0.61 | -5.57 | -6.57 | -0.19 |
| | [19,26] | **-12.26** | -5.54 | +2.59 | -4.37 | **-5.32** | **+0.77** | -5.48 | **-6.54** | +0.77 |
| | Merged | -9.66 | -2.23 | **+3.92** | **-7.99** | -2.80 | -0.79 | **-8.80** | -1.24 | **+1.87** |
| Shikra-7B | [3,13] | -5.46 | -3.05 | -0.06 | **-10.69** | -4.06 | +2.43 | -1.96 | -8.27 | +0.69 |
| | [14,28] | **-10.41** | -5.13 | +0.90 | -9.26 | **-8.94** | +1.29 | **-8.11** | -5.13 | +0.33 |
| | Merged | -2.17 | **-8.67** | **+1.91** | -10.30 | -6.13 | **+3.34** | -6.32 | **-10.86** | **+3.44** |

**Table 2:** Relative improvement (%) on hallucination metrics for LLaVA-7B under random attribution and hallucination-focused attribution. **Bold numbers** indicate the best performance for each metric within each group and split.

| Model | Group | Validation Δ (%) | | | Hallucination Δ (%) | | | Generalization Δ (%) | | |
|---|---|---|---|---|---|---|---|---|---|---|
| | | CHAIR$_s$ | CHAIR$_i$ | F1 | CHAIR$_s$ | CHAIR$_i$ | F1 | CHAIR$_s$ | CHAIR$_i$ | F1 |
| LLaVA-7B | [5,18] | -6.85 | -1.96 | +0.60 | **-11.32** | -3.81 | **+1.31** | -4.11 | -5.72 | +0.37 |
| | [19,26] | -8.58 | -7.27 | +1.35 | -5.24 | -3.16 | +0.73 | -3.61 | -5.02 | +1.63 |
| | Merged | -8.05 | -3.29 | -0.04 | -9.77 | -4.34 | -0.07 | -6.25 | -5.55 | +0.55 |
| LLaVA-7B-hallu | [5,18] | -9.03 | -6.12 | **+3.20** | -7.55 | -1.35 | +0.70 | -11.03 | -14.12 | +1.41 |
| | [19,26] | -3.18 | -1.49 | +1.97 | -3.87 | +1.36 | +0.63 | -9.60 | -2.81 | +1.01 |
| | Merged | **-10.90** | **-7.40** | +2.19 | -8.32 | **-9.13** | +0.57 | **-15.27** | **-19.98** | **+3.36** |

pipeline in Section 4.3, this analysis focuses on the *consistency* and *stability* of pruning effects. The central questions are: (1) Can our attribution and pruning strategy consistently suppress hallucination across diverse model architectures? (2) Does this suppression generalize to out-of-distribution data splits?

We construct three evaluation sets for each model:

- **Validation Set:** Images used for attribution in each head group (160 per group for LLaVA-7B/13B; 320 for Shikra-7B). For merged groups, we take the union of primary groups after removing duplicates.

- **Hallucination Set:** Images that trigger fully hallucinatory captions (CHAIR$_s = 1$) under baseline inference; for each group, the number of images is matched to its Validation Set.

- **Generalization Set:** Images outside the Validation Set, randomly sampled to the same size as each group's Validation Set to enable evaluation under distribution shift.

Following the empirical findings of Li et al. (2025), for each model, we define three attribution layer groups: LLaVA uses [5,18], [19,26], and a merged [5,26]; Shikra-7B uses [3,13], [14,28], and a merged [3,28]. The merged group unites non-overlapping intervals to enhance coverage. Faithful and hallucinatory heads are identified via a multi-round attribution procedure for statistical reliability. Grid search for hyperparameters ($\mu$, $\lambda$) is conducted independently on each split, with settings detailed in Appendix C.

As shown in Table 1, our attribution-guided pruning consistently reduces both CHAIR$_s$ and CHAIR$_i$ across all models and evaluation splits, demonstrating stable suppression of hallucinations under varying architectural and distributional conditions. Notably, F1 remains stable or improves in nearly all cases, underscoring the robustness and generalizability of our method in mitigating hallucinations without sacrificing informativeness.

**Table 3:** Comparison of our method with hallucination-mitigation methods on hallucination-prone evaluation sets for each model. For each **model block**, best results are in **bold** and second best are underlined (excluding baseline).

| Model | Method | CHAIR$_s$ | CHAIR$_i$ | Recall | Precision | F1 | Len |
|---|---|---|---|---|---|---|---|
| LLaVA-7B | Baseline | 1.000 | 0.320 | 0.792 | 0.585 | 0.673 | 92.24 |
| | VCD | 0.731 | 0.259 | 0.760 | 0.643 | 0.697 | 92.33 |
| | OPERA | 0.731 | 0.234 | **0.827** | 0.683 | 0.748 | 95.99 |
| | PAI | **0.450** | **0.155** | 0.647 | **0.778** | 0.707 | 78.26 |
| | MLIH | 0.637 | 0.211 | 0.759 | 0.709 | 0.733 | 80.01 |
| | SPIN | 0.807 | 0.255 | 0.817 | 0.637 | 0.716 | 100.07 |
| | VHR | 0.525 | 0.221 | 0.679 | 0.698 | 0.689 | 64.83 |
| | Ours | 0.494 | 0.158 | 0.772 | 0.760 | **0.766** | 83.94 |
| Shikra-7B | Baseline | 0.785 | 0.245 | 0.778 | 0.623 | 0.692 | 99.97 |
| | OPERA | 0.690 | 0.232 | 0.760 | 0.670 | 0.710 | 101.20 |
| | PAI | 0.715 | 0.227 | 0.780 | 0.643 | 0.705 | 98.14 |
| | MLIH | **0.335** | **0.123** | 0.546 | **0.800** | 0.649 | 53.20 |
| | SPIN | 0.570 | 0.178 | 0.729 | 0.694 | 0.711 | 92.61 |
| | VHR | 0.540 | 0.198 | 0.662 | 0.709 | 0.684 | 75.20 |
| | Ours | 0.601 | 0.161 | **0.789** | 0.725 | **0.755** | 98.89 |
| Qwen2.5-VL-7B-Instruct | Baseline | 0.694 | 0.216 | 0.512 | 0.750 | 0.609 | 120.14 |
| | VCD | 0.744 | 0.254 | 0.488 | 0.730 | 0.585 | 110.86 |
| | OPERA | 0.556 | 0.255 | 0.338 | 0.718 | 0.460 | 119.75 |
| | PAI | 0.563 | 0.244 | 0.381 | 0.747 | 0.504 | 88.82 |
| | MLIH | **0.431** | 0.236 | 0.284 | 0.733 | 0.410 | 63.54 |
| | SPIN | 0.481 | 0.230 | 0.284 | 0.761 | 0.414 | 62.86 |
| | VHR | 0.525 | 0.201 | 0.404 | 0.780 | 0.532 | 105.43 |
| | Ours | 0.600 | **0.148** | **0.545** | **0.817** | **0.654** | 120.43 |

### 4.3 Hallucination-Focused Attribution: An Empirical Exploration

Hallucination cases provide attribution signals that are rarely visible in ordinary validation images, exposing larger variations in attention patterns and InfoSpectralScore responses. This motivates an alternative setting in which attribution is computed on intentionally challenging inputs, with the expectation that such cases may better reveal heads associated with hallucination and thus support more effective pruning. Concretely, for each model, we identify images that produce hallucinated objects under baseline inference and use these hallucination-prone cases as challenging attribution inputs. For LLaVA, hallucination-prone cases are generally reproducible; for Shikra and Qwen2.5, the exact set varies slightly across runs due to decoding stochasticity and model-specific generation behavior. Therefore, we treat these challenging examples as a model-dependent source of high-risk inputs rather than a fixed dataset.

**Finding 1: Hallucination-focused attribution provides stronger and more stable pruning signals.** We compare attributions computed from random images and hallucination-prone images under the same generalization pipeline and multi-round aggregation strategy. As shown in Table 2 for LLaVA-7B, hallucination-focused attribution consistently yields larger reductions in both CHAIR$_s$ and CHAIR$_i$ across validation, hallucination, and out-of-distribution splits. Among all layer groups, the merged group exhibits the most pronounced improvements. While these results do not imply causal interpretations of head behavior, they empirically indicate that high-risk cases expose more informative head-level attribution patterns, enabling more robust pruning.

**Finding 2: Hallucination-focused attribution enhances task-specific hallucination mitigation.** Building on these observations, we incorporate attribution derived from hallucination-prone cases into our full task-specific pruning pipeline, which performs instance-level head reweighting and pruning at inference. As shown in Table 3, using hallucination-focused attribution as the head-importance prior substantially improves hallucination suppression for LLaVA-7B, Shikra-7B, and Qwen2.5-VL-7B-Instruct on their respective hallucination-prone evaluation sets. Our method achieves the lowest or second-lowest CHAIR$_i$ across all three models while maintaining competitive recall and F1. In contrast, several baselines suppress hallucinations at the cost of caption length or substantial recall degradation. These findings demonstrate that attribution signals extracted from challenging cases offer effective

**Table 4:** Comparison of hallucination-mitigation methods on a dataset of 600 randomly selected multi-label COCO images (each with $\geq 5$ labels). Best results are in **bold** and second best are underlined.

| Method | CHAIR$_s$ | CHAIR$_i$ | Recall | Precision | F1 | Len |
|---|---|---|---|---|---|---|
| Baseline | 0.678 | 0.184 | 0.695 | 0.762 | 0.727 | 105.202 |
| VCD | 0.678 | 0.177 | **0.721** | 0.763 | 0.741 | 115.647 |
| OPERA | 0.663 | 0.177 | 0.720 | 0.775 | **0.747** | 114.343 |
| PAI | **0.410** | **0.121** | 0.504 | **0.841** | 0.630 | 82.922 |
| MLIH | 0.619 | 0.165 | 0.654 | 0.781 | 0.712 | 94.913 |
| SPIN | 0.494 | 0.147 | 0.529 | 0.802 | 0.637 | 71.942 |
| VHR | 0.700 | 0.190 | 0.709 | 0.752 | 0.730 | 108.830 |
| Ours | 0.532 | 0.130 | 0.683 | 0.820 | 0.745 | 102.972 |

**Table 5:** Comparison of hallucination-mitigation methods on the full POPE benchmark. Best results are in **bold** and second best are underlined.

| Method | Random | | | | Popular | | | | Adversarial | | | |
|---|---|---|---|---|---|---|---|---|---|---|---|---|
| | Acc | Prec | Rec | F1 | Acc | Prec | Rec | F1 | Acc | Prec | Rec | F1 |
| Baseline | 88.47 | 95.72 | 80.53 | 87.47 | 86.83 | 92.14 | 80.53 | 85.95 | 83.50 | 85.61 | 80.53 | 83.00 |
| VCD | 85.83 | 93.59 | 76.93 | 84.45 | 84.40 | 90.44 | 76.93 | 83.14 | 81.33 | 84.36 | 76.93 | 80.47 |
| OPERA | 89.23 | 92.55 | 85.33 | 88.80 | 86.83 | 87.97 | 85.33 | 86.63 | 81.17 | 78.77 | 85.33 | 81.92 |
| PAI | 88.27 | 85.56 | 92.07 | 88.70 | 84.87 | 80.62 | 91.80 | 85.85 | 76.53 | 70.33 | 91.80 | 79.64 |
| MLIH | 81.10 | **98.95** | 62.87 | 76.89 | 80.73 | **97.82** | 62.87 | 76.54 | 79.87 | **95.25** | 62.87 | 75.74 |
| SPIN | 89.57 | 94.26 | 84.27 | 88.98 | 87.10 | 89.38 | 84.20 | 86.71 | 83.23 | 82.31 | 84.67 | 83.47 |
| VHR | 88.87 | 86.30 | **92.40** | 89.25 | 85.37 | 81.01 | **92.40** | 86.33 | 77.23 | 70.90 | **92.40** | 80.23 |
| Ours | **90.40** | 98.56 | 82.00 | **89.52** | **89.83** | 97.38 | 81.87 | **88.95** | **88.73** | 94.97 | 81.80 | **87.89** |

guidance for task-specific head intervention, yielding robust improvements over existing training-free approaches. Qualitative examples in Appendix D further illustrate this behavior.

## 4.4 Broad Evaluation Across Diverse Hallucination Benchmarks

The previous sections examined attribution behavior and high-risk hallucination scenarios. We now evaluate whether the task-specific pruning pipeline generalizes to broader distributions and task forms. To this end, we assess performance across four representative benchmarks: a multi-label subset of COCO, the full POPE benchmark, HallusionBench, and MME. These datasets jointly cover caption-level hallucination, object-level binary consistency, visual illusion diagnosis, and multimodal perception.

**COCO Multi-Label Subset: Real-World Captioning** Table 4 reports results on a randomly selected set of 600 multi-label COCO images, where captions must describe numerous co-occurring objects. In this setting, several methods such as VCD, OPERA, and VHR bring limited reductions in hallucination, suggesting that their behaviors do not transfer as effectively to complex multi-object scenes. Other approaches, including PAI and SPIN, suppress hallucinations more aggressively but often at the cost of shorter outputs and reduced recall. Our method achieves a more balanced outcome: it reduces hallucinated content while maintaining caption length and sustaining strong recall, leading to competitive overall F1 performance. These results indicate that the proposed intervention adapts well to dense visual scenes without pushing the model toward overly conservative descriptions.

**Table 6:** Correctness comparison on HallusionBench with LLaVA-1.5-7B under human evaluation. All numbers are in %.

| Method | Question Pair Accuracy | Figure Accuracy | Easy Accuracy | Hard Accuracy | Overall Accuracy |
|---|---|---|---|---|---|
| Baseline | 17.36 | 19.36 | 30.55 | 57.67 | 53.14 |
| Ours | **17.80** | **19.94** | **32.53** | 57.67 | **54.03** |

**Table 7:** Results on MME perception-related tasks. Best values are **bolded**.

| | Existence | Count | Position | Color | Posters | Celebrity | Scene | Landmark | Artwork | OCR | Total |
|---|---|---|---|---|---|---|---|---|---|---|---|
| Baseline | 185.00 | 138.33 | 108.33 | 155.00 | 152.04 | 130.29 | 163.00 | 143.00 | 102.00 | 97.50 | 1374.50 |
| Ours | **190.00** | **145.00** | **120.00** | 155.00 | **184.69** | **147.94** | **179.00** | **146.75** | **116.00** | **102.50** | **1486.89** |

**Table 8:** Results on MME recognition-related tasks. Best values in each column are **bolded**.

| | Common Sense Reasoning | Numerical Calculation | Text Translation | Code Reasoning | Total |
|---|---|---|---|---|---|
| Baseline | 102.14 | 60.00 | 87.50 | 55.00 | 304.64 |
| Ours | **114.29** | **90.00** | 87.50 | **72.50** | **364.29** |

**POPE: Object-Level Binary Consistency**    Table 5 presents results on the full POPE benchmark covering the Random, Popular, and Adversarial splits. In all three settings, our method demonstrates stable improvements in object-level consistency while maintaining balanced precision and recall. These results indicate that the proposed head-level adjustment preserves the model's ability to make reliable object-present/object-absent decisions even under distributional variation.

**HallusionBench: Visual Illusion and Reasoning Robustness**    HallusionBench includes visual illusions, ambiguous scenes, and paired reasoning questions that challenge even stronger LVLMs. As shown in Table 6, our method yields only modest improvements over the baseline. This limited gain likely reflects the difficulty of the benchmark and the limited influence that head-level intervention can exert on high-level reasoning tasks. Even so, the intervention remains stable and yields small, consistent improvements without affecting reasoning behavior.

**MME: Multimodal Perception and Cognitive Evaluation**    MME covers a broad range of perception and reasoning abilities beyond hallucination. As shown in Tables 7 and 8, our method generally maintains or improves performance across most perception-oriented tasks, indicating that hallucination reduction does not weaken visual grounding. Tasks that rely on concrete visual cues, such as object presence, counting, or basic scene interpretation, tend to show clearer improvements. This outcome is consistent with our goal of mitigating functional conflicts within attention heads: reducing spurious visual–text correlations helps stabilize perception- oriented behavior. Tasks involving abstraction or symbolic manipulation show smaller changes, consistent with their weaker connection to hallucination-related behavior.

Overall, these results show that the task-specific pruning pipeline remains consistent across diverse benchmarks, improving hallucination control while preserving perceptual grounding and caption informativeness.

## 5    Conclusion

This work presents a unified, training-free framework for fine-grained control of object hallucination in vision–language models. By introducing InfoSpectralScore for head-level attribution and a dynamic counteractive pruning strategy, our method suppresses hallucination-prone heads while reinforcing faithful ones, without requiring extra data or retraining. Experiments across LVLMs, together with benchmarks such as POPE, show consistent hallucination reduction while preserving informativeness, providing both strong empirical results and mechanistic insights into the role of attention heads.

Looking ahead, an exciting avenue is to extend our head-level paradigm beyond attention modules to other critical components such as MLP layers, and to design more efficient mechanisms for intervening in key computational units. Moreover, future work may explore more expressive attribution scores and algorithmic strategies that further enhance the efficiency, scalability, and interpretability of hallucination control in multimodal generation.

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

APPENDIX

# A    COMPREHENSIVE EXPERIMENTAL CONFIGURATION AND PIPELINE-LEVEL ENHANCEMENTS

## A.1    EXPERIMENTAL SETTINGS

### METHOD-SPECIFIC CONFIGURATION NOTES

All experiments are conducted under identical hardware and fixed random seeds.

**OPERA.**   We strictly follow the official implementation and its reported best-practice hyperparameters. The command lines used in our experiments correspond exactly to those recommended in the authors' repository, with all other parameters kept at their default values.

**SPIN.**   For LLaVA-1.5 on CHAIR evaluation, the configuration reported in the SPIN paper (`start-layer`=0, `end-layer`=32) remains stable and we use it without modification. However, applying the same configuration to POPE causes severe degradation of generation quality. The original paper provides no task-specific guidance, and we therefore explored several alternative layer ranges. Among these tested configurations, we report results using `start-layer`=5 and `end-layer`=32, which consistently yielded the most stable and competitive performance on POPE. A similar phenomenon is observed for Shikra, where the 0–32 intervention led to unusable outputs. Following controlled exploration, we adopted the configuration used in our final results: `start-layer`=5, `end-layer`=32, `routed-heads`=0.45, `small-num-mask`=0.001.

**PAI.**   For LLaVA-1.5 on the CHAIR benchmark, we follow the configuration described in the original paper, using interventions on layers 2–32 together with `alpha`=0.5, `gamma`=1.1, and classifier-free attention. On POPE, however, applying the same layer span leads to noticeable instability. Because the paper does not provide a recommended setting for POPE, we carried out a moderate search over several candidate layer ranges and chose layers 6–20, which showed the most reliable performance among the tested options. All other hyperparameters remain aligned with the CHAIR configuration.

**VCD.**   We follow the hyperparameters reported in the official paper. Specifically, we use $\gamma = 0.1$, $\alpha = 1.0$, and $\beta = 0.1$, while the number of noise steps is set to $T = 500$ for MME and LLaVA-Bench, and to $T = 999$ for POPE. All remaining parameters are kept exactly as in the authors' codebase.

**VHR.**   All experiments with VHR strictly follow the repository-provided scripts and default hyperparameters for both POPE and CHAIR, without modification.

**MLIH.**   We reproduce MLIH using the hyperparameters provided in the paper: for LLaVA-1.5, we use `guide-range`=5–18 and `alpha`=0.5; for Shikra, we use `guide-range`=3–13 and `alpha`=0.55. These intervention ranges naturally align with those used in our own method, ensuring fair and architecture-consistent comparison.

### ADJUSTMENTS FOR QWEN2.5-VL-INSTRUCT

Compared with LLaVA and Shikra, Qwen2.5-VL-Instruct exhibits stronger baseline expressiveness and noticeably more unstable decoding under loosely constrained prompts or layer-level interventions. To ensure fair comparison and generate usable captions across all SOTA methods, we introduce the following minimal and model-specific adjustments.

**Prompt adaptation.**   Most models (e.g., LLaVA and Shikra) are evaluated using the default prompt: "Please describe the image in detail." However, Qwen2.5-VL-Instruct exhibits noticeably more unstable generations under this loose instruction, often producing overly

long, drifting, or structurally irregular text. To stabilize decoding and maintain comparability across methods, we adopt the following controlled prompt:

> "Write one fluent paragraph of about 100–120 English words that thoroughly describes the image. Mention the main subjects, their attributes, spatial relations, actions, visible text or numbers, and the overall scene context. Avoid generic prefaces such as 'The image shows', as well as bullet points or headings. The prose should remain cohesive and natural, and the description must end with `<END>`."

Several SOTA methods fail to produce valid text under this strict prompt (e.g., empty outputs, corrupted tokens, premature EOS). In such cases, we apply only the minimal relaxation needed to recover valid captions, for example by removing the sentence "Mention the main subjects, their attributes, spatial relations, actions, visible text or numbers, and the overall scene context." or by omitting the `<END>` marker, while still keeping the overall prompt structure and evaluation conditions as consistent as possible.

**Architectural alignment and layer mapping.** Several competing methods specify intervention layers only for LLaVA (e.g., layers 2–31). To preserve comparable relative depth, we map these ranges to Qwen2.5 by aligning proportional layer indices (e.g., 2–27). For methods that operate on middle layers (ours and MLIH), we follow prior analyses of Qwen architectures (HALP Anonymous (2025), Qwen-LA Chu et al. (2025)) and adopt:

$$\texttt{TARGET\_LAYERS} = [12, 13, 14, 15, 16, 18, 19, 20, 22, 27],$$

which covers layers 12–16, 18–22, and 27. Layer 21 is excluded from modification due to instability but retained as a probing layer for token extraction.

**Other adjustments.** All remaining hyperparameters (decoding strategy, max tokens, safety checks, and batch configuration) are kept identical to those used for LLaVA and Shikra, unless a method fails to run under the common configuration. In such rare cases, we introduce the smallest possible fix required for successful execution without changing the method's effective behavior.

## A.2 Key Engineering Improvements for the Task-Specific Pruning Pipeline

### Accelerated Head-Level Attribution via Cached Semantic Reconstruction

A direct implementation of Algorithm 1 would require running the LVLM once for every head ablation in every target layer, resulting in hundreds of full forward passes per attribution set. This is prohibitively expensive for modern LVLMs. To address this, we adopt a fast approximation that performs *only one* baseline forward pass and caches, for each target layer:

- the residual stream before attention,
- the concatenated multi-head output *before the output projection $W_o$*,
- the post-attention semantic embedding of the final token.

Because the post-attention update for each head is linear before the post-attention layer normalization, the effect of ablating a head can be reconstructed offline. For a cached layer $l$, let $\mathbf{h}_j^{(l)} \in \mathbb{R}^{d_h}$ be the output of head $j$, and let $W_o^{(l)}$ be the output projection. The reconstructed semantic embedding under ablating head $j$ is

$$\tilde{\mathbf{x}}_{\text{sem}}^{(l)} = \mathbf{x}_{\text{resid}}^{(l)} + W_o^{(l)}\big(\text{concat}(\mathbf{h}_1^{(l)}, \ldots, \mathbf{0}, \ldots, \mathbf{h}_H^{(l)})\big), \tag{13}$$

followed by the same post-attention layer normalization used in the model. This enables the computation of InfoSpectralScore for all heads via Eqs. (4)–(6) without invoking additional forward passes or generating captions. In practice, this approximation shrinks the attribution time from minutes-to-hours down to mere seconds, effectively rendering the cost negligible and enabling exhaustive head-by-head and layer-by-layer analysis on every model.

**Why this approximation is valid.** The key observation is that the final-token semantic vector before the MLP block is a linear function of the concatenated head outputs:

$$\mathbf{x}_{\text{sem}}^{(l)} = \text{LN}\Big(\mathbf{x}_{\text{resid}}^{(l)} + W_o^{(l)} \text{concat}(\mathbf{h}_1^{(l)}, \ldots, \mathbf{h}_H^{(l)})\Big). \tag{14}$$

Thus, modifying only one head corresponds to a low-rank perturbation in the semantic space. Since InfoSpectralScore depends solely on the covariance structure of these semantic embeddings, we can compute the change in spectral statistics directly from reconstructed embeddings.

Empirically, we observed that this fast approximation yields head-importance rankings that closely track those produced by the full forward-regeneration procedure. More importantly, when incorporated into the broader analyses of attribution behavior and hallucination scenarios presented in Sections 4.3 and 4.4, the approximation continues to support effective hallucination mitigation. These empirical trends suggest that the cached-semantic representation preserves the discriminative signals required for reliable head attribution, while making instance-level adaptation computationally feasible in practice.

Stabilized and Efficient Bayesian Optimization for Per-Instance Pruning

The task-specific pruning pipeline (Algorithm 3) uses a lightweight Bayesian optimization (BO) loop to determine the pruning parameters $(\mu, \lambda)$ for each input instance. Although the search space is only two-dimensional, each evaluation requires full LVLM decoding, making runtime the primary computational bottleneck. To ensure that BO remains both stable and efficient in this setting, we introduce several practical refinements.

**Warm-start initialization combined with random exploration.** Instead of relying solely on uninformed random initialization, we supply BO with a small set of empirically strong configurations gathered from preliminary development runs. These warm-start seeds are combined with a few genuinely uninformed points, balancing guided initialization with exploratory diversity. This mixture places the optimizer near effective regions of the pruning space while still allowing it to probe areas that were not covered during development.

**Constrained search region.** We restrict the optimization domain to

$$\mu \in [0.3, 0.8], \qquad \lambda \in [0.80, 0.99],$$

removing degenerate settings that either oversuppress faithful heads ($\mu$ too large) or destabilize modulation sensitivity ($\lambda$ too close to 1). The constrained region retains sufficient expressivity while substantially reducing the variance of early BO iterations.

**Robust handling of irregular LVLM generations.** Under our dynamic pruning method, LVLM decoding may occasionally become unstable and produce malformed outputs, such as truncated sentences, incoherent token sequences, or repeated prefix fragments. If these outputs were treated as valid optimization samples, they could distort the BO surrogate model and impede stable convergence. To avoid this, we apply lightweight validity checks that examine output length, character-level anomalies, and prefix-overlap patterns. Outputs that fail these checks receive a soft penalty rather than being removed, which keeps the surrogate model reliable while maintaining smooth optimization progress for each instance.

**Early stopping based on difficult-to-improve hallucination gains.** Extensive empirical analysis shows that some inputs yield consistently non-positive hallucination improvements across a broad range of $(\mu, \lambda)$ settings. This behavior typically arises when the pruning adjustment causes unstable decoding, producing borderline or partially degraded text, or when the objective surface becomes effectively flat with respect to the pruning parameters and no meaningful improvement can be reached. In these situations, continued exploration brings little benefit. Therefore, the BO loop is terminated early once several consecutive iterations show no improvement or fall below a small gain threshold, preventing excessive and unproductive evaluations on instances that are intrinsically difficult to optimize.

**Table A1:** Runtime on CHAIR evaluation with 600 COCO multi-label images. All time entries are in **hh:mm:ss**.

| Method | Baseline | SPIN | PAI | OPERA | VCD | VHR | MLIH | Ours |
|---|---|---|---|---|---|---|---|---|
| Total Time | 00:17:49 | 00:20:30 | 00:39:52 | 03:10:02 | 00:39:10 | 00:20:10 | 00:16:18 | 09:50:00 |
| Avg Time (s/img) | 1.98 | 2.05 | 3.99 | 19.00 | 3.92 | 2.02 | 1.63 | 148.00 |

**Table A2:** Runtime on POPE evaluation. The upper block reports runtime on the three POPE subsets (random / popular / adversarial). The lower block reports the total runtime and the average time per question (total time divided by 9000). All time entries are in **hh:mm:ss**.

| Method | Baseline | SPIN | PAI | OPERA | VCD | VHR | MLIH | Ours |
|---|---|---|---|---|---|---|---|---|
| **Subset Runtime** | | | | | | | | |
| Random | 00:13:17 | 00:06:05 | 00:22:19 | 01:04:57 | 00:08:17 | 00:09:42 | 00:10:57 | 01:07:04 |
| Popular | 00:13:21 | 00:06:55 | 00:21:48 | 01:06:30 | 00:08:16 | 00:09:40 | 00:10:55 | 01:08:52 |
| Adversarial | 00:14:09 | 00:06:17 | 00:21:27 | 01:06:10 | 00:08:17 | 00:09:41 | 00:11:21 | 01:13:21 |
| **Overall Runtime** | | | | | | | | |
| Total Time | 00:40:48 | 00:19:17 | 01:05:34 | 03:17:37 | 00:24:50 | 00:29:03 | 00:33:13 | 03:29:17 |
| Avg Time (s/question) | 0.272 | 0.128 | 0.437 | 1.317 | 0.166 | 0.194 | 0.221 | 1.395 |

**Fallback retry for atypical attribution distributions.** If the primary BO phase fails to identify an improving configuration, we activate a short retry stage. This stage recomputes hallucination heads using a robust interquartile-range (IQR) rule and runs a small BO pass with freshly initialized priors. The fallback mechanism compensates for unusually flat attribution distributions and ensures that the pruning pipeline remains reliable even on challenging inputs.

With these refinements, a full optimization cycle typically completes within 3–5 minutes. Given that head-level pruning inherently incurs additional decoding passes, this runtime remains practical for per-instance hallucination mitigation.

### A.3 COMPUTATIONAL COST

Tables A1 and A2 summarize the end-to-end runtime of all competing methods on CHAIR (600 captions, max_new_tokens=256) and POPE (9000 binary questions, max_new_tokens=16). Our method introduces two additional computations per instance: (i) head-level attribution, and (ii) the Bayesian search for $(\mu, \lambda)$ during counteractive pruning.

**Attribution cost.** As defined in Algorithm 1, attribution requires evaluating each attention head on an attribution set of size $K$. With the cached reconstruction mechanism (Appendix A.2), the entire attribution step becomes lightweight:

- **5 seconds per instance** on CHAIR (long captions);
- **0.5 seconds per instance** on POPE (short answers).

**Overall runtime and scalability.** The attribution and pruning components operate strictly on a per-instance basis and therefore do not scale with the size of the evaluation set: once a single image or question has been processed, its associated computational cost does not accumulate across larger datasets.

On CHAIR, long-form caption generation (256 tokens) naturally increases latency, since counteractive pruning requires several forward passes under different $(\mu, \lambda)$ configurations. Although this makes our method slower than other training-free modulation techniques on this benchmark, the incurred cost remains within a practically acceptable range for the practical usage of the approach, especially in scenarios where maintaining semantic richness and performing per-instance hallucination mitigation are both required.

For POPE, which involves short answers (16 tokens), our method remains within the same order of magnitude as all compared approaches. Even with the additional attribution step (about 0.5 s per instance), the overall end-to-end runtime stays moderate and does not introduce prohibitive overhead. These results indicate that the method is suitable for real applications requiring adaptive, instance-level hallucination control.

## B  HYPERPARAMETER SENSITIVITY ANALYSIS

**Table A3:** Grid search results for $\alpha$ and $\gamma$ (showing $\Delta_h$, $\Delta_f$, and Signal). Signal uses the `signal` value. The top-3 Signal is in bold.

| $\alpha$ | $\gamma$ | $\Delta_h$ | $\Delta_f$ | Signal |
|---|---|---|---|---|
| 0.0 | 0.0 | 0.047041 | 0.146614 | 0.193655 |
| 0.0 | 0.5 | 0.033169 | 0.148833 | 0.182002 |
| 0.0 | 1.0 | 0.033169 | 0.148833 | 0.182002 |
| 0.0 | 1.5 | 0.033169 | 0.148833 | 0.182002 |
| 0.0 | 2.0 | 0.033169 | 0.148833 | 0.182002 |
| 0.1 | 0.0 | 0.032024 | 0.110746 | 0.142770 |
| 0.1 | 0.5 | 0.027930 | 0.194834 | 0.222764 |
| 0.1 | 1.0 | 0.037277 | 0.149162 | 0.186438 |
| 0.1 | 1.5 | 0.033169 | 0.149162 | 0.182331 |
| 0.1 | 2.0 | 0.033169 | 0.149162 | 0.182331 |
| 0.2 | 0.0 | 0.032024 | 0.110923 | 0.142948 |
| 0.2 | 0.5 | 0.039011 | 0.194834 | 0.233845 |
| 0.2 | 1.0 | 0.027930 | 0.194834 | 0.222764 |
| 0.2 | 1.5 | 0.037277 | 0.149162 | 0.186438 |
| 0.2 | 2.0 | 0.037277 | 0.149162 | 0.186438 |
| 0.3 | 0.0 | 0.032024 | 0.110923 | 0.142948 |
| 0.3 | 0.5 | 0.025653 | 0.196326 | 0.221979 |
| 0.3 | 1.0 | 0.067022 | 0.194834 | **0.261855** |
| 0.3 | 1.5 | 0.027930 | 0.194834 | 0.222764 |
| 0.3 | 2.0 | 0.025941 | 0.195190 | 0.221131 |
| 0.4 | 0.0 | 0.032024 | 0.110923 | 0.142948 |
| 0.4 | 0.5 | 0.025653 | 0.185017 | 0.210670 |
| 0.4 | 1.0 | 0.039011 | 0.194834 | 0.233845 |
| 0.4 | 1.5 | 0.027930 | 0.194834 | 0.222764 |
| 0.4 | 2.0 | 0.027930 | 0.194834 | 0.222764 |
| 0.5 | 0.0 | 0.032024 | 0.110923 | 0.142948 |
| 0.5 | 0.5 | 0.072374 | 0.185625 | **0.257999** |
| 0.5 | 1.0 | 0.070548 | 0.185123 | **0.255672** |
| 0.5 | 1.5 | 0.039011 | 0.194834 | 0.233845 |
| 0.5 | 2.0 | 0.027930 | 0.194834 | 0.222764 |

Due to the absence of ground-truth annotations for identifying hallucination-inducing or faithful attention heads, we design a proxy evaluation framework using CHAIR metrics. Specifically, we assess the semantic alignment improvement brought by pruning heads selected under different InfoSpectralScore regularization weights.

To validate the reliability of this proxy experiment, we conduct systematic studies on the LLaVA-13B model, focusing on layers 5 through 7. The experimental protocol is as follows:

1. **Sample Preparation:** We first perform inference on 300 samples using the original model, and utilize an automatic CHAIR-based evaluator to obtain 170 faithful samples and 130 hallucinated samples. From these, we randomly select 32 faithful and 32 hallucinated samples as evaluation subsets for hyperparameter analysis.

2. **Grid Search and Consistency Scoring:** We conduct a grid search over InfoSpectralScore regularization weights $\alpha \in \{0.0, 0.1, ..., 0.5\}$ and $\gamma \in \{0.0, 0.5, ..., 2.0\}$. For each configuration, we compute the InfoSpectralScore-based consistency scores for all attention heads in the selected layers, aggregating over 15 stochastic generations per head.

3. **Head Selection via Quantile Strategy:** For each layer, hallucination heads are selected from both ends, and faithful heads from the center, of the consistency score distribution.

4. **Head Pruning and CHAIR Evaluation:** For each head group, we prune the selected heads in each layer (e.g., 20 hallucination heads and 10 faithful heads per layer, out of 40 total), and evaluate the model's performance before and after counteractive pruning using the CHAIR-i and CHAIR-s metrics, separately on the hallucinated and faithful samples:

$$bh_i, bh_s, \quad bf_i, bf_s, \quad ph_i, ph_s, \quad pf_i, pf_s$$

where $bh_i, bh_s$ denote baseline CHAIR-i/s scores on hallucinated samples, $bf_i, bf_s$ on faithful samples, $ph_i, ph_s$ after pruning hallucination heads (on hallucinated samples), and $pf_i, pf_s$ after pruning faithful heads (on faithful samples).

5. **Pruning Effectiveness Signal:** The pruning effectiveness is measured by the signal:

$$\Delta_h = (bh_i - ph_i) + (bh_s - ph_s) \tag{15}$$
$$\Delta_f = (bf_i - pf_i) + (bf_s - pf_s) \tag{16}$$
$$\text{Signal} = \Delta_h + \Delta_f \tag{17}$$

where a higher Signal indicates more effective and selective pruning.

6. **Hyperparameter Selection:** Rather than selecting a single optimum, we adopt the top-3 $(\alpha, \gamma)$ configurations (with the highest Signal values) as candidate hyperparameters for all subsequent evaluations.

Based on the grid search results in Table A3, we identify a clear optimal region in the $(\alpha, \gamma)$ hyperparameter space. The top-3 configurations—$(0.3, 1.0)$, $(0.5, 0.5)$, and $(0.5, 1.0)$—achieve the highest Signal values ($0.2619$, $0.2580$, and $0.2557$, respectively), substantially outperforming the remaining settings.

Across all hyperparameter combinations, the Signal metric remains positive (ranging from approximately $0.1428$ to $0.2619$), indicating that our InfoSpectralScore-based counteractive pruning reliably distinguishes between hallucinatory and faithful heads in an unsupervised manner. Notably, the improvement on hallucinated samples ($\Delta_{\text{hall}}$) ranges from $0.0257$ to $0.0724$, while the effect on faithful samples ($\Delta_{\text{faith}}$) ranges from $0.1107$ to $0.1963$; the top-3 configurations maximize both metrics simultaneously.

**Parameter comparison:** The influence of $\alpha$ and $\gamma$ on the two key metrics exhibits distinct patterns. Increasing $\alpha$ (the spectral variance penalty) leads to a direct and substantial increase in $\Delta_{\text{hall}}$—for example, when $\gamma$ is fixed, raising $\alpha$ from $0.0$ to $0.5$ elevates $\Delta_{\text{hall}}$ from approximately $0.033$ to $0.072$. In contrast, $\gamma$ (the spectral entropy weighting) primarily affects $\Delta_{\text{faith}}$, but this effect is relatively moderate: for fixed $\alpha$, varying $\gamma$ results in only minor changes in $\Delta_{\text{faith}}$ (typically within the range $0.1107$–$0.1963$). This demonstrates that the method is particularly sensitive to the choice of $\alpha$ for hallucination suppression, while being robust to $\gamma$ in terms of preserving performance on faithful samples.

In summary, we adopt the top-3 hyperparameter groups as candidate settings in subsequent experiments, ensuring our method operates in the most robust and effective region identified by quantitative analysis of the search space.

## C    Hyperparameter Validation for Counterbalance

To validate the robustness of our dynamic counterbalance mechanism, we conduct an extensive hyperparameter search over the modulation strength $\mu$ and smoothing factor $\lambda$, which govern the magnitude and temporal stability of per-head activation adjustment.

**Experimental Setup.**    Experiments are performed on the LLaVA-7B model, with dynamic counteractive pruning applied to layers 5 through 18 and baseline. Predefined sets of hallucination-inducing and faithful heads—obtained via attribution on a dedicated 160-image

**Table A4:** Line search over modulation strength $\mu$ (with fixed $\lambda = 0.95$) for dynamic counterbalance. Reported values are the relative improvement (%) on CHAIR-s, CHAIR-i, and F1.

| $\mu$ | $\lambda$ | CHAIR-s (%) | CHAIR-i (%) | F1 (%) |
|---|---|---|---|---|
| 0.20 | 0.95 | $-2.70$ | $-0.84$ | 0.49 |
| 0.25 | 0.95 | 0.00 | $-1.30$ | 0.74 |
| 0.30 | 0.95 | $-4.05$ | $-6.94$ | 1.74 |
| 0.35 | 0.95 | 1.35 | $-5.20$ | 0.93 |
| 0.40 | 0.95 | 2.70 | $-3.63$ | $-0.41$ |
| 0.45 | 0.95 | 0.00 | $-8.50$ | 0.90 |
| 0.50 | 0.95 | $-2.70$ | $-9.22$ | 0.52 |
| 0.55 | 0.95 | 0.00 | $-7.53$ | $-0.03$ |
| 0.60 | 0.95 | 0.00 | $-9.25$ | 1.13 |
| 0.65 | 0.95 | 0.00 | $-8.15$ | 0.36 |

**Table A5:** Grid search over modulation strength $\mu$ and smoothing factor $\lambda$ for dynamic counterbalance. Values show relative improvement (%) on CHAIR-s, CHAIR-i, and F1.

| $\mu$ | $\lambda$ | CHAIR-s (%) | CHAIR-i (%) | F1 (%) |
|---|---|---|---|---|
| 0.30 | 0.93 | $-1.35$ | $-1.16$ | 1.38 |
| 0.30 | 0.95 | $-4.05$ | $-5.66$ | 1.42 |
| 0.30 | 0.97 | $-5.41$ | $-4.00$ | 1.34 |
| 0.30 | 0.99 | $-2.70$ | $-2.48$ | 1.21 |
| 0.30 | 1.00 | $-1.35$ | $-1.22$ | 1.23 |
| 0.50 | 0.93 | $-2.70$ | $-9.34$ | 0.60 |
| 0.50 | 0.95 | $-2.70$ | $-9.98$ | 0.51 |
| 0.50 | 0.97 | $-2.70$ | $-9.51$ | 0.67 |
| 0.50 | 0.99 | $-1.35$ | $-7.87$ | 0.09 |
| 0.50 | 1.00 | $-1.35$ | $-7.55$ | 0.15 |
| 0.55 | 0.93 | 0.00 | $-8.62$ | $-0.19$ |
| 0.55 | 0.95 | 0.00 | $-7.57$ | 0.03 |
| 0.55 | 0.97 | 0.00 | $-8.52$ | 0.40 |
| 0.55 | 0.99 | $-1.35$ | $-9.99$ | 0.11 |
| 0.55 | 1.00 | $-1.35$ | $-8.54$ | $-0.16$ |
| 0.60 | 0.93 | 0.00 | $-9.34$ | 0.28 |
| 0.60 | 0.95 | 0.00 | $-11.03$ | 0.89 |
| 0.60 | 0.97 | 0.00 | $-11.37$ | 0.83 |
| 0.60 | 0.99 | 0.00 | $-11.61$ | 0.61 |
| 0.60 | 1.00 | 0.00 | $-11.13$ | 0.98 |
| 0.65 | 0.93 | 0.00 | $-9.87$ | 1.13 |
| 0.65 | 0.95 | 0.00 | $-8.20$ | 0.63 |
| 0.65 | 0.97 | $-1.35$ | $-12.87$ | 1.13 |
| 0.65 | 0.99 | $-2.70$ | $-12.83$ | 1.43 |
| 0.65 | 1.00 | 0.00 | $-11.02$ | 1.00 |

**Table A6:** Best hyperparameter settings for each model, group, and evaluation set.

| Model | Group | Evaluation Set | $\mu$ | $\lambda$ |
|---|---|---|---|---|
| LLaVA-7B | [5,18] | Validation | 0.60 | 0.99 |
| LLaVA-7B | [5,18] | Hallucination | 0.65 | 0.95 |
| LLaVA-7B | [5,18] | Generalization | 0.65 | 0.95 |
| LLaVA-7B | [19,26] | Validation | 0.65 | 0.90 |
| LLaVA-7B | [19,26] | Hallucination | 0.65 | 0.99 |
| LLaVA-7B | [19,26] | Generalization | 0.65 | 0.97 |
| LLaVA-7B | Merged Set | Validation | 0.62 / 0.65 | 0.80 / 0.97 |
| LLaVA-7B | Merged Set | Hallucination | 0.60 / 0.70 | 0.99 / 0.90 |
| LLaVA-7B | Merged Set | Generalization | 0.62 / 0.65 | 0.97 / 0.97 |
| LLaVA-7B-hallu | [5,18] | Validation | 0.60 | 0.95 |
| LLaVA-7B-hallu | [5,18] | Hallucination | 0.60 | 0.95 |
| LLaVA-7B-hallu | [5,18] | Generalization | 0.65 | 0.90 |
| LLaVA-7B-hallu | [19,26] | Validation | 0.65 | 0.99 |
| LLaVA-7B-hallu | [19,26] | Hallucination | 0.60 | 0.99 |
| LLaVA-7B-hallu | [19,26] | Generalization | 0.35 | 0.97 |
| LLaVA-7B-hallu | Merged Set | Validation | 0.65 / 0.60 | 0.90 / 0.99 |
| LLaVA-7B-hallu | Merged Set | Hallucination | 0.65 / 0.65 | 0.95 / 0.97 |
| LLaVA-7B-hallu | Merged Set | Generalization | 0.60 / 0.60 | 0.90 / 0.99 |
| LLaVA-13B | [5,18] | Validation | 0.65 | 0.97 |
| LLaVA-13B | [5,18] | Hallucination | 0.60 | 0.85 |
| LLaVA-13B | [5,18] | Generalization | 0.55 | 0.97 |
| LLaVA-13B | [19,26] | Validation | 0.50 | 0.99 |
| LLaVA-13B | [19,26] | Hallucination | 0.65 | 0.97 |
| LLaVA-13B | [19,26] | Generalization | 0.55 | 0.80 |
| LLaVA-13B | Merged Set | Validation | 0.50 / 0.99 | 0.90 / 0.97 |
| LLaVA-13B | Merged Set | Hallucination | 0.60 / 0.65 | 0.85 / 0.80 |
| LLaVA-13B | Merged Set | Generalization | 0.60 / 0.65 | 0.80 / 0.80 |
| Shikra-7B | [3,13] | Validation | 0.30 | 0.95 |
| Shikra-7B | [3,13] | Hallucination | 0.75 | 0.90 |
| Shikra-7B | [3,13] | Generalization | 0.67 | 0.90 |
| Shikra-7B | [14,28] | Validation | 0.60 | 0.80 |
| Shikra-7B | [14,28] | Hallucination | 0.75 | 0.99 |
| Shikra-7B | [14,28] | Generalization | 0.60 | 0.90 |
| Shikra-7B | Merged Set | Validation | 0.30 / 0.75 | 0.95 / 0.97 |
| Shikra-7B | Merged Set | Hallucination | 0.67 / 0.60 | 0.90 / 0.80 |
| Shikra-7B | Merged Set | Generalization | 0.75 / 0.60 | 0.90 / 0.90 |

dataset—are fixed for all runs. Counteractive pruning and evaluation are directly performed on this set to ensure a consistent basis for hyperparameter validation.

We first fix $\lambda = 0.95$ and perform a line search over $\mu \in [0.2, 0.65]$ (step size 0.05), as shown in Table A4. Subsequently, we perform a two-dimensional grid search with $\mu \in \{0.3, 0.5, 0.55, 0.60, 0.65\}$ and $\lambda \in \{0.93, 0.95, 0.97, 0.99, 1.0\}$ (see Table A5). Each setting is evaluated using the hook-based intervention (see Section 3.2), and CHAIR-s, CHAIR-i, and F1 metrics are reported as the relative improvement (%) over the baseline.

**Results.** Our results demonstrate that appropriate tuning of $\mu$ and $\lambda$ can yield substantial improvements in both hallucination mitigation and overall task performance. Notably, ($\mu = 0.3, \lambda = 0.97$) achieves the largest reduction in CHAIR-s ($-5.41\%$) among all settings, while ($\mu = 0.60, \lambda = 0.95$) achieves the largest reduction in CHAIR-i ($-11.03\%$) and also improves F1. These findings validate the effectiveness and flexibility of the counterbalance mechanism and provide practical guidance for selecting robust hyperparameter combinations in future deployments.

**Selection Across All Model Groups.** For completeness, the optimal settings of $(\mu, \lambda)$ for each model, group, and evaluation split are provided in Table A6. These values are determined based on the grid search results above and subsequent validation experiments across the full model suite.

# D  QUALITATIVE CASE STUDY

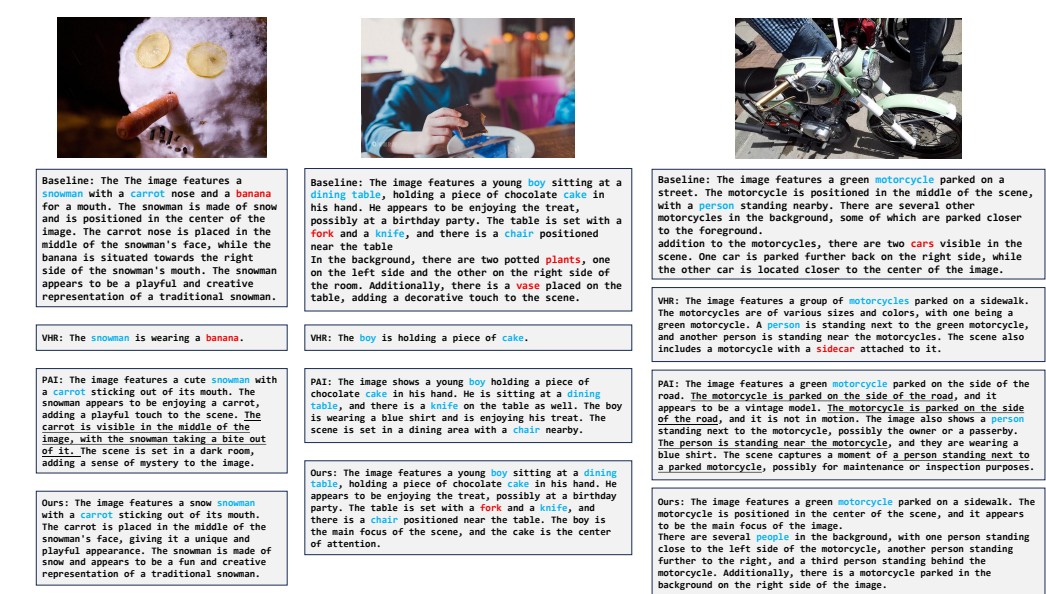

**Figure A1:** Qualitative results of hallucination mitigation on LLaVA-1.5-7B. Real and hallucinated object words are marked in blue and red, respectively. The prompt is "Please describe the image in detail."

To provide a more intuitive comparison among our approach (HACP), VHR, and PAI, we present qualitative captioning examples generated by the LLaVA-1.5-7B model. Consistent with Section 4.3, these examples are selected from a curated set of 160 COCO validation images that consistently induce severe hallucinations (CHAIR-s = 1) under baseline inference.

Figure A1 shows three representative cases, comparing captions from the baseline, VHR, PAI, and our method. Real object mentions are highlighted in blue, and hallucinated objects in red. The generation prompt was: *"Please describe the image in detail."*

The first two examples demonstrate that captions produced by VHR exhibit an especially sharp reduction in length compared to other methods, often omitting many relevant details. In addition, in the third example, which features a relatively complex scene, VHR still generates the shortest and least informative caption among all methods. This pattern clearly illustrates VHR's overly conservative tendency, resulting in both minimal length and reduced content richness even for images that would otherwise support more expressive descriptions.

Additionally, the captions generated by PAI in Examples 1 and 3 display structurally repetitive patterns and content homogeneity, reflecting the phenomenon of "structurally uniform, less diverse outputs" observed previously. Such repetition diminishes the informativeness and engagement of the captions.

In contrast, our method (HACP) effectively reduces hallucinations while preserving richer and more expressive content. The generated captions remain detailed, maintaining high coverage of correct object mentions. This highlights the superior balance our approach achieves between hallucination suppression and caption informativeness.

# E  THE USE OF LARGE LANGUAGE MODELS (LLMs)

In this work, large language models (LLMs) were used to assist with writing. Their role was limited to polishing sentences, clarifying ambiguous expressions, and maintaining a

scientific, rigorous, and efficient writing style. LLMs were not involved in research ideation, experimental design, data analysis, or the development of our proposed methods. The authors take full responsibility for all content presented in this paper.

