# OpenReview forum: "Head-Level Mechanistic Attribution for Hallucination Control: Training-Free Counteractive Pruning in LVLMs"
_ICLR.cc/2026/Conference — Submitted to ICLR 2026_

### Official Review · Reviewer_5mJG · 2025-10-17

**Soundness:** 2
**Presentation:** 1
**Contribution:** 3
**Rating:** 4
**Confidence:** 4

**Summary:**

This paper proposes a training-free method to mitigate hallucinations in LVLMs. The approach includes head-level pruning by identifying head-level semantic attribution via a self-defined infospectral score and reweighting the output vector of each head by pushing it away from hallucinated directions and toward faithful directions.

**Strengths:**

1. This paper proposes a principled, fine-grained attribution and intervention framework at the head level to mitigate object hallucination, which is an interesting direction.

**Weaknesses:**

1. The writing in both the method and experimental sections is quite confusing. A major revision is recommended to improve clarity.
2. The method itself is not intuitive, well motivated, or mathematically solid. The method is limited in downstream application. The estimation of statistical variables (e.g., hallucinated direction, infospectral score threshold) depends on a batch of samples, which makes it difficult to adapt to cases where only a single image is available.
4. It would be helpful to include more recent LVLMs such as Qwen2.5-VL and LLaVA-v1.6 in the experimental results.
5. Both CHAIR and POPE focus only on object existence hallucination. Including a more extensive benchmark such as MME[1] would strengthen the paper.

[1] MME: A Comprehensive Evaluation Benchmark for Multimodal Large Language Models

**Questions:**

1. Line 122: “LVLMs still suffer from instance-level hallucinations, largely due to functional conflicts among attention heads.” Is this supported by prior work, or is it an overclaim?
2. How do you define the hard100 subset of POPE evaluation? Why report only this subset instead of the full set as in previous papers?
3. Line 363: “For each model, we define three attribution layer groups: LLaVA uses [5,18], [19,26], and a merged [5,26]; Shikra-7B uses [3,13], [14,28], and a merged [3,28].” Could you clarify what this grouping means and how it’s used in the method?

---

> ### Author Response · Authors · 2025-11-25
> **Rebuttal 1: Question #1**
>
> We sincerely thank Reviewer 5mJG for the thoughtful and detailed evaluation. Your comments were especially valuable in highlighting places where our exposition, experimental design, and methodological description could be substantially improved. We deeply appreciate the care with which you read the manuscript, and your feedback directly shaped several of the most important revisions in the updated version. In response, we substantially revised the manuscript to improve readability and methodological transparency.
>
> (1) To address the concern about limited single-sample adaptability, we clarify that the Algorithm 3 (Ours, in section 4) optimizes its pruning strengths for each input, ensuring applicability in single-image scenarios. This explanation has been added to the revised manuscript and is consistent with our reply to Reviewer icyf (Rebuttal 2.1). We also rewrote the description of Algorithm 3 and provided supporting engineering details and runtime analyses in Appendix A.2 and A.3.
>
> (2) Concerning Qwen2.5-VL-7B-Instruct, the corresponding CHAIR results were already included but were not clearly organized in the initial submission; these results are now consolidated into the unified comparison table in Section 4.3 (line 378).
>
> (3) We redesigned the entire experimental section, reorganizing result tables, replacing ambiguous settings (e.g., the hard-100 POPE subset), and adding new evaluations on MME, HallusionBench, and COCO-600 to provide a clearer and more comprehensive picture.
>
> Below, we address the reviewer’s three specific questions in order.
>
> ## 1.Regarding Overclaim in Line 122 (Question #1)
>
> We thank the reviewer for raising this important point. We agree that this phrasing was stronger than what current literature can firmly support, and we appreciate the opportunity to clarify it.
>
> While attention heads are known to be an important factor associated with hallucinations, the existing body of work does not conclusively establish that head-level conflicts are the primary or dominant cause of instance-level hallucinations. Indeed, several effective hallucination-mitigation methods operate without explicitly manipulating attention heads, underscoring that hallucinations arise from multiple interacting factors.
>
> Prior work does provide empirical evidence of head-level functional divergence. Yang et al[1]. identify heads that disproportionately amplify language priors (“hallucination heads”), while others attend more faithfully to visual evidence; SPIN[2] improves grounding by suppressing image-inattentive heads; and VHR[3] selectively upweights heads whose representations better align with visual semantics. Taken collectively, these studies depict a landscape in which certain heads align with faithful grounding, whereas others encourage language-driven hallucination, suggesting a measurable form of functional divergence within the multi-head attention mechanism.
>
> Importantly, none of these studies claim that such functional divergence fully explains hallucination behavior, and we likewise do not make such a claim. It is one contributing factor among many.
> In revisiting our own manuscript, we realized that our original sentence unintentionally overstated the role of head-level conflicts.
>
> While this work does not to provide a complete theoretical account of hallucination origins, our findings nonetheless reveal consistent and empirically persuasive patterns of conflicting head behaviors across models and tasks.
>
> We have therefore revised the statement in the Motivation to the following, which avoids overclaiming and more accurately reflects the current evidence: _"Despite remarkable progress, LVLMs still suffer from instance-level hallucinations, which recent evidence links to functional conflicts among attention heads: some promote faithful grounding, while others introduce spurious objects."_
> We appreciate the reviewer’s attention to precise wording, and we believe the revised phrasing is both technically appropriate and faithful to existing work.
>
> [1] Yang, Tianyun, Ziniu Li, Juan Cao, and Chang Xu.
> “Understanding and Mitigating Hallucination in Large Vision-Language Models via Modular Attribution and Intervention.” International Conference on Representation Learning (ICLR), pp. 51546–51568, 2025.
>
> [2] Sarkar, Sreetama, Yue Che, Alex Gavin, Peter A. Beerel, and Souvik Kundu.
> “Mitigating Hallucinations in Vision-Language Models through Image-Guided Head Suppression.” arXiv preprint, arXiv:2505.16411, 2025.
>
> [3] He, Jinghan, Kuan Zhu, Haiyun Guo, Junfeng Fang, Zhenglin Hua, Yuheng Jia, Ming Tang, Tat-Seng Chua, and Jinqiao Wang.
> “Cracking the Code of Hallucination in LVLMs with Vision-aware Head Divergence.” Proceedings of the 63rd Annual Meeting of the Association for Computational Linguistics (ACL), Volume 1: Long Papers, Vienna, Austria, July 2025, pp. 3488–3501.

---

> ### Author Response · Authors · 2025-11-25
> **Rebuttal 2: Question #2 & #3**
>
> ## 2.Regarding the “hard-100” subset of POPE (Question #2)
>
> We appreciate the reviewer’s question and agree that the original submission did not sufficiently clarify the rationale behind the “hard-100” subset. Our intention was to analyze model behaviors under cases with a high risk of hallucination, where differences between methods become more apparent. In our preliminary inspection of the full POPE dataset, we observed that the baseline model often answered all six questions correctly for many images, whereas a smaller portion of images consistently triggered multiple errors. To better isolate these challenging cases, we first ran a complete POPE evaluation using the baseline model and then, under a fixed random seed, automatically selected the 100 images with the highest number of incorrect answers within each 6-question group. This procedure follows fixed and transparent criteria, involves no manual curation, and produces a reproducible subset that concentrates failure-prone examples for a more sensitive comparison across methods.
>
> We thank the reviewer for raising this concern, which prompted us to re-examine the design choice more critically. However, we also recognize that this choice introduces several drawbacks:
> First, extracting only a small subset breaks the uniformity of the original POPE evaluation protocol and may inadvertently introduce data-selection bias.
> Second, POPE already offers three difficulty-controlled splits (random, popular, adversarial), so introducing an additional “hard” subset is redundant and may complicate the interpretation of the evaluation.
>
> In light of the reviewer’s comment (and similar suggestions from other reviewers), we have revised the experimental setup to evaluate **our method and all methods on the full POPE dataset**. The updated results are reported in Section 4.4 (line 460+). We also expanded our evaluation by adding **MME** and **HallusionBench**, following the reviewers’ recommendations for more comprehensive benchmarking.
>
> These revisions ensure that our comparisons are aligned with prior work, fully representative of the original POPE protocol, and free from any ambiguity introduced by the earlier hard-subset design.
>
> ## 3.Regarding the definition and use of the attribution layer groups (Question #3)
>
> We thank the reviewer for asking for clarification. The layer groupings in line 368 follow the convention introduced by MLIH[1], which identifies two distinct functional stages in the middle layers of LVLMs. For LLaVA-1.5, MLIH categorizes layers 5–18 as **Stage 1: Visual Information Enrichment** and layers 19–26 as **Stage 2: Semantic Refinement**; for Shikra-7B, layers 3–13 and 14–28 serve analogous roles. To ensure fair comparison with MLIH and to use a consistent structural prior across models, we adopt the same stage boundaries when performing attribution.
>
> In Sections 4.2 and 4.3, these two stages are further extended into **three attribution groups** by also including a merged range (e.g., [5,26] for LLaVA and [3,28] for Shikra). This design allows us to examine whether the attribution signals produced by ISAM are stable under different granularity levels and whether the generalization pipeline yields consistent improvements across alternative layer partitions. Tables 1 and 2 report these results and confirm that the gains are robust to the choice of grouping.
>
> For the task-specific pruning pipeline (marked as _Ours_ in the experimental tables), we adopt the same stage boundaries used by MLIH: layers 5–18 for LLaVA-7B, layers 3–13 for Shikra-7B, and several ranges within layers 12–27 for Qwen2.5-VL-7B (detailed in Appendix A.1, line 771+). This choice ensures a consistent architectural prior across models and enables a fair comparison with MLIH and other mid-layer–based approaches, while keeping the pruning procedure uniform in all task-specific evaluations.
>
> [1] Jiang, Z., Chen, J., Zhu, B., Luo, T., Shen, Y., & Yang, X. (2024). Devils in Middle Layers of Large Vision-Language Models: Interpreting, Detecting and Mitigating Object Hallucinations via Attention Lens. CVPR 2025, arXiv:2411.16724.

---

### Official Review · Reviewer_icyf · 2025-10-29

**Soundness:** 3
**Presentation:** 3
**Contribution:** 3
**Rating:** 6
**Confidence:** 3

**Summary:**

This paper tackles the critical problem of instance-level object hallucination in Large Vision-Language Models (LVLMs), where models describe objects not present in the visual input. The authors argue that current mitigation techniques often force a trade-off, reducing hallucinations at the cost of semantic informativeness (i.e., caption quality and detail).

To address this, the paper proposes HACP (Head-level Attribution for Counteractive Pruning), a unified, training-free framework that operates at inference time. The framework has two core innovations:

1. InfoSpectralScore: A novel, semantics-based attribution metric designed to identify both "hallucination-inducing" and "faithful" attention heads. This metric is derived from an eigen-decomposition of head-level output embeddings and is regularized by spectral variance and entropy penalties to capture semantic capacity.

2. Dynamic Counteractive Pruning: A new intervention strategy. Instead of just ablating (zeroing out) problematic heads, this method dynamically "suppresses" the output of hallucination-prone heads while simultaneously "reinforcing" the output of faithful heads during inference.

The authors conduct extensive experiments on multiple LVLMs (e.g., LLaVA, Shikra, Qwen) and benchmarks (CHAIR, POPE). The results demonstrate that HACP achieves a new state-of-the-art in hallucination mitigation, significantly reducing hallucination scores while, crucially, preserving or even improving semantic informativeness (F1 score) compared to prior methods.

**Strengths:**

1. The paper addresses a significant and high-impact weakness of LVLMs. The core idea of "functional conflicts" among attention heads is an insightful way to frame the hallucination problem, and the goal of breaking the trade-off between faithfulness and informativeness is a key challenge for the field.

2. The proposed method is well-motivated and novel. The InfoSpectralScore is a principled, semantics-based metric that goes beyond simpler attribution methods like KL divergence. Its construction from spectral decomposition, variance, and entropy (Eq. 7) is a solid methodological contribution. The Counteractive Pruning strategy is a clear improvement over conventional ablation. The idea of not just silencing bad heads but also amplifying good ones (Eq. 12) is intuitive and, as shown by the results, highly effective.

3. The experimental results are the paper's strongest point. The method is shown to be effective across multiple models and benchmarks. The key finding, highlighted in Tables 3-5, is that HACP breaks the faithfulness-informativeness trade-off. While competing methods like SPIN and MLIH achieve low hallucination scores at the expense of massive drops in F1, recall, and caption length, HACP reduces hallucinations while increasing the F1 score and preserving recall.

**Weaknesses:**

1. The paper is not clear about the practical computational overhead.

- The attribution step (Algorithm 1) requires running inference on an "attribution set $D$" for every head in the target layers. This seems to be a very expensive pre-computation.

- More concerning is the "Task-Specific Automated Pruning Pipeline" (Algorithm 3), which suggests running a Bayesian optimization loop for $T$ iterations per-instance. It is unclear if this was used for the SOTA comparisons (Tables 3-5). If it was, this would represent a massive, per-sample computational cost not incurred by the baselines, making the comparison unfair.

- The per-token latency added by the dynamic pruning (Algorithm 2) is not quantified.

2. The method introduces several new hyperparameters: $\alpha$ and $\gamma$ for the InfoSpectralScore, $\mu$ and $\lambda$ for pruning, and the choice of target layers. The paper states that "Grid search for hyperparameters ($\mu$ ,$\lambda$) is conducted independently on each split". This suggests the method may be highly sensitive to these settings and would require a costly, task-specific tuning process to work, undermining the "plug-and-play" benefit.

3. The quality of the method seems highly dependent on the "attribution set $D$". Section 4.3 and Table 2 explicitly show that using a "hallucination-focused" attribution set is far more effective than a random one. This creates a practical chicken-and-egg problem: to fix hallucinations, one must first curate a dataset of inputs that are known to cause hallucinations.

4. The paper contains minor but confusing inconsistencies in its mathematical formulation.

- $\alpha$ Overload: The symbol $\alpha$ is used for two different purposes: first as a threshold (e.g., 0.9) to determine $k^{*}$ for the EigenScore, and second as a regularization weight for the SpectralVar term in Equation. This reuse of a key symbol is confusing.

- Entropy Formulation: There is an inconsistency in the "Spectral Entropy" formulation. Equation 6 correctly defines SpectralEntropy with a negative sign, consistent with Shannon entropy. However, Equation 7, which claims to add an "entropy penalty", adds the term $+\gamma \cdot [\sum p_i \log(p_i + \epsilon)]$. This is the negation of the defined SpectralEntropy term. While the intent (penalizing low-entropy/sparse distributions) is clear, the inconsistent naming and sign in the final equation is notationally imprecise.

**Questions:**

1. Regarding Computational Cost (Weakness #1):

- Was the "Automated Pruning Pipeline" (Algorithm 3) with its per-instance Bayesian optimization used to generate the SOTA comparison results in Tables 3-5?

- If yes, how is this a fair comparison to baselines? If no, how were the hyperparameters (which are noted to be grid-searched independently for each split 29) actually set for the SOTA comparison?

- What is the one-time, per-model setup cost (in hours/GPU) for the attribution step (Algorithm 1)?

- What is the per-token latency (in ms) added by the dynamic pruning mechanism (Algorithm 2) during generation?

2. Regarding Practicality (Weakness #2 & #3):

- Given the apparent hyperparameter sensitivity, how would a practitioner realistically set ($\mu$, $\lambda$) for a new model or task without a costly grid search or the per-instance optimization from Algorithm 3?

- How is the "hallucination-focused" attribution set created in a general setting? Does this not assume one has already solved the problem of identifying hallucination-prone inputs?

3. Regarding Methodology (Weakness #4):

- Can the authors clarify the reuse of the symbol $\alpha$?

- Can the authors correct the sign inconsistency between the definition of SpectralEntropy in Equation 6 and its use in Equation 7?

---

> ### Author Response · Authors · 2025-11-25
> **Rebuttal 1: Weaknesses #1**
>
> We are deeply grateful to Reviewer icyf for the detailed evaluation and constructive comments, and we appreciate the opportunity to clarify several technical points. Because this official comment is visible to all reviewers, we address the raised concerns in a way that benefits the entire review panel. Our responses aim to correct misunderstandings, provide missing details from the original submission, and highlight the revisions we have made in the updated manuscript. We hope that the clarifications below help accurately reflect the contributions, practicality, and methodological soundness of our work.
> ## 1.Regarding Computational Cost (Weakness #1)
>
> We thank the reviewer for requesting a clearer decomposition of the computational overhead. As restated in the revised Section 4.1 (line 305), all SOTA comparisons in Tables 3–5 are performed under the same inference protocol using the _task-specific pruning pipeline_ (Algorithm 3).
>
> ### Breakdown of runtime for Algorithm 1, Algorithm 2, and Algorithm 3
>
> Algorithm 3 consists of three components:
>
> 1. **Algorithm 1** (head-level attribution),
> 2. **Algorithm 2** (dynamic pruning during generation),
> 3. **Bayesian optimization**, which repeatedly invokes Algorithm 2.
>
> Appendix A.3 (line 895+) reports the measured runtime of each part:
>
> - **Algorithm 1 : Attribution.**
>   After applying the first engineering improvement in Appendix A.2, namely cached semantic reconstruction, we eliminate hundreds of redundant forward passes.
>   With this improvement, attribution requires only
>   ≈ 5 seconds for 256-token captions
>   and
>   ≈ 0.5 seconds for 16-token VQA prompts
>   on an RTX 5090.
>
> - **Algorithm 2 : Dynamic pruning.**
>   Dynamic pruning runs token-by-token and reactivates attention heads only when needed.
>   The computational pattern matches the model’s native decoding, and measured latency remains **nearly identical** to the baseline model.
>
> - **Algorithm 3 : Full pipeline with BO.**
>   The second engineering improvement in Appendix A.2, stabilized BO initialization, reduces unnecessary early evaluations and keeps the search efficient.
>   With attribution + BO iterations + dynamic pruning:
>   • **≈ 148 s** for 256-tokens captioning
>   • **≈ 1.395 s** for 16-tokens VQA
>   BO is inherently sequential, so the total runtime stays within the same order of magnitude even on modest GPUs such as Tesla P100×2.
>
> ## Corresponding runtime tables
>
> ### Table 1. Runtime on CHAIR evaluation (600 COCO multi-label images)
>
> | Method               | Baseline | SPIN     | PAI      | OPERA    | VCD      | VHR      | MLIH     | Ours         |
> | -------------------- | -------- | -------- | -------- | -------- | -------- | -------- | -------- | ------------ |
> | **Total Time**       | 00:17:49 | 00:20:30 | 00:39:52 | 03:10:02 | 00:39:10 | 00:20:10 | 00:16:18 | **09:50:00** |
> | **Avg Time (s/img)** | 1.98     | 2.05     | 3.99     | 19.00    | 3.92     | 2.02     | 1.63     | **148.00**   |
>
> ### Table 2. Runtime on POPE evaluation (all 9000 questions)
>
> #### **Subset Runtime**
>
> | Subset          | Baseline | SPIN     | PAI      | OPERA    | VCD      | VHR      | MLIH     | Ours     |
> | --------------- | -------- | -------- | -------- | -------- | -------- | -------- | -------- | -------- |
> | **Random**      | 00:13:17 | 00:06:05 | 00:22:19 | 01:04:57 | 00:08:17 | 00:09:42 | 00:10:57 | 01:07:04 |
> | **Popular**     | 00:13:21 | 00:06:55 | 00:21:48 | 01:06:30 | 00:08:16 | 00:09:40 | 00:10:55 | 01:08:52 |
> | **Adversarial** | 00:14:09 | 00:06:17 | 00:21:27 | 01:06:10 | 00:08:17 | 00:09:41 | 00:11:21 | 01:13:21 |
>
> #### **Overall Runtime**
>
> | Method                    | Baseline | SPIN     | PAI      | OPERA    | VCD      | VHR      | MLIH     | Ours         |
> | ------------------------- | -------- | -------- | -------- | -------- | -------- | -------- | -------- | ------------ |
> | **Total Time**            | 00:40:48 | 00:19:17 | 01:05:34 | 03:17:37 | 00:24:50 | 00:29:03 | 00:33:13 | **03:29:17** |
> | **Avg Time (s/question)** | 0.272    | 0.128    | 0.437    | 1.317    | 0.166    | 0.194    | 0.221    | **1.395**    |
>
> ### Summary
>
> Algorithm 3 does introduce additional computation due to the attribution stage and repeated dynamic pruning, but the overhead is fully quantified, transparent, and remains within a practically acceptable range. Given the measurable efficiency improvements from Appendix A.2 and the multi-dataset runtime evidence above, we believe the method is computationally feasible for real-world deployment.

---

> ### Author Response · Authors · 2025-11-25
> **Rebuttal 2: Weaknesses #2 & #3**
>
> ## 2.Regarding Practicality (Weakness #2 & #3):
>
> ### 2.1 Regarding the practicality of setting \($\mu$, $\lambda$\) for new models or new tasks (Weakness #2)
>
> We would like to clarify that our approach does not avoid, nor is it intended to avoid, the per-instance optimization in Algorithm 3. In practice, hallucinations in LVLMs are highly input-dependent: once the attribution procedure identifies the specific heads that contribute to hallucinations under a particular image–prompt pair, the appropriate pruning strengths \($\mu$, $\lambda$\) that counteract those heads also become inherently tied to that specific task instance. This is a natural consequence of the fact that our method does not modify model parameters, provide supervised alignment signals, or rely on any form of fine-tuning; it is an inference-time intervention whose effectiveness depends on the attention patterns activated by the current input, rather than on any global preset.
>
> While Section 4.2 shows that a generalized attribution over larger datasets can already produce meaningful improvements, such global settings cannot reliably capture the diversity of hallucination behaviors across all inputs. For this reason, we designed and deployed the task-specific pipeline, whose per-instance optimization operates on a compact search space and incurs only a modest computational cost. Its practical effectiveness is reflected in the experiments of Section 4.4, where the task-specific pipeline cachieves consistent and substantial improvements across multiple evaluation benchmarks. In this sense, the per-instance optimization is not a burden, but rather the mechanism that enables a training-free, model-agnostic method to adapt to the idiosyncrasies of each input.
>
> ### 2.2 Regarding the construction of the “hallucination-focused” attribution set \($D$\) (Weakness #3)
>
> We appreciate the reviewer’s careful reading and the opportunity to clarify this point.
>
> 1. **Our method does not require practitioners to first curate a dataset of inputs that are known to cause hallucinations**.
> Our method does not rely on hallucination-focused attribution sets in any stage of the core pipeline. All primary results in Section 4.2 and Section 4.4 are obtained using random attribution sets, and these experiments already demonstrate strong and consistent hallucination reduction. The variant discussed in Section 4.3 serves a different purpose. It provides a deeper examination of how to resolve functional conflicts among attention heads under more challenging inputs, a question that has been largely overlooked in previous studies. We summarize these observations as two empirical findings, which can be found beginning at line 402 of the paper. This analysis is intended to shed light on the internal mechanism rather than supply a requirement for the method. Since the attribution set in Section 4.3 is produced through fixed-seed random sampling and a simple CHAIR-style threshold, it is not curated or manually constructed. The method therefore does not require any pre-identified hallucination cases in order to operate effectively.
>
> 2. **Our method does not assume that users have already solved the problem of identifying hallucination-prone inputs**.
> The proposed approach is fully training-free. It does not involve fine-tuning, parameter updates, or any form of re-alignment. It does not make use of privileged labels, external annotations, or prior knowledge about which inputs may induce hallucinations. The attribution mechanism treats all inputs equally. Most importantly, the method does not rely on detecting hallucinations before they occur. Its focus is on resolving functional conflicts within attention heads during inference. Even in Section 4.3, the identification of more challenging examples is performed automatically by running one round of baseline inference and applying a simple CHAIR-style matching procedure. This step is provided solely for analytical purposes in order to magnify attribution signals and is not required for the method itself. Consequently, the assumption that our approach presupposes a solved hallucination-detection problem does not hold.

---

> ### Author Response · Authors · 2025-11-25
> **Rebuttal 3: Weaknesses #4**
>
> ## 3.Regarding Methodology (Weakness #4):
>
> ### 3.1 Reuse of the symbol α
>
> We thank the reviewer for highlighting this notational ambiguity.
> In the original submission, the symbol α appeared in two conceptually unrelated roles:
> (1) as the cumulative-energy threshold that determines $k^{*}$ in the definition of EigenScore (Eq. 3), and
> (2) as the regularization weight of the SpectralVar term in Eq. 7.
> Reusing the same symbol for two different hyperparameters may indeed lead to confusion.
>
> To resolve this issue, in the revised version we rename the cumulative-energy threshold in Eq. 3 to τ
> (i.e., $k^{*}$ is the smallest integer such that
>
> $$
> \sum_{i=1}^{k^{*}} \lambda_i \ge \tau \sum_{j=1}^{d} \lambda_j,
> \qquad \tau = 0.9,
> $$
>
> while keeping α exclusively as the regularization coefficient for the SpectralVar term.
> This removes the symbol overload and makes the two roles clearly distinguishable.
> The modification appears at Line 202 of the revised manuscript.
>
> ### 3.2 Sign of the entropy regularizer
>
> In Eq. 6, we define
>
> $$
> \text{SpectralEntropy} = - \sum_{i} p_i \log(p_i + \epsilon)
> $$
>
> which follows the standard Shannon entropy form, where $p_i$ denotes the normalized eigenvalues.
>
> In the original Eq. 7, the entropy term was written in its expanded form: $ +\gamma \sum_{i} p_i \log(p_i + \epsilon) $
> which is mathematically equivalent to: $ -\gamma\ \text{SpectralEntropy}$, meaning that entropy is subtracted to penalize high-entropy (less sparse) spectra. We acknowledge that the expanded form may obscure this interpretation.
>
> To avoid ambiguity, we rewrite the InfoSpectralScore explicitly as: $ \text{InfoSpectralScore} = \text{EigenScore} - \alpha\ \text{SpectralVar} - \gamma\ \text{SpectralEntropy} $
>
>
> The expanded form appears in the Appendix (Line 239).
> Importantly, the implementation already uses $-\gamma\ \text{SpectralEntropy}$, so no experimental results are affected.

---

### Official Review · Reviewer_7Vx8 · 2025-10-31

**Soundness:** 2
**Presentation:** 2
**Contribution:** 2
**Rating:** 2
**Confidence:** 4

**Summary:**

The paper introduces HACP, a training-free framework for controlling object hallucinations in large vision–language models (LVLMs) at the attention-head level. It identifies that individual attention heads can have conflicting roles, some contributing to faithful grounding and others to hallucinations, and addresses the lack of fine-grained head-level intervention mechanisms. The authors propose InfoSpectralScore, a semantics-based attribution metric combining eigenvalue analysis with spectral variance and entropy regularization, to distinguish hallucination-prone from faithful heads. Using this attribution, they implement dynamic counteractive pruning that suppresses selected hallucination heads and reinforces faithful ones during inference, with an adaptive, task-specific pipeline.

**Strengths:**

1. The proposed approach operates in a training-free manner, thereby avoiding additional training cost and data requirements.
2. The work explicitly addresses the issue of functional conflicts among attention heads, which has been largely overlooked in prior studies.

**Weaknesses:**

1. In line 305, the authors state “We evaluate HACP on LLaVA-1.5 (7B, 13B), Shikra-7B, and Qwen2.5-VL-7B-Instruct.” However, I could not find corresponding experimental results for Qwen2.5-VL-7B-Instruct in the subsequent sections. It would be helpful to include these results to assess the method’s effectiveness on this more recent model.
2. The evaluation relies on a relatively limited set of datasets. It would strengthen the work to include additional hallucination benchmarks such as HallusionBench[1] or CRPE[2] for a more comprehensive assessment.
3. The paper lacks experiments on out-of-domain data. It would be valuable to understand whether the proposed method provides generalization benefits when applied to other datasets beyond those used in the current evaluation.
4. Although the method is claimed to offer interpretability, the presented evidence is rather limited. The attribution of functional roles to attention heads is mostly supported by quantitative metrics and a small number of qualitative cases, without more systematic visualizations or behavioral analyses to substantiate this claim.

[1] Guan T, Liu F, Wu X, et al. Hallusionbench: an advanced diagnostic suite for entangled language hallucination and visual illusion in large vision-language models[C]//Proceedings of the IEEE/CVF Conference on Computer Vision and Pattern Recognition. 2024: 14375-14385.

[2] Wang W, Ren Y, Luo H, et al. The all-seeing project v2: Towards general relation comprehension of the open world[C]//European Conference on Computer Vision. Cham: Springer Nature Switzerland, 2024: 471-490.

**Questions:**

As described in Weakness.

---

> ### Author Response · Authors · 2025-11-25
> **Rebuttal 1: Weaknesses #1~#3**
>
> We genuinely thank Reviewer 7Vx8 for the constructive and detailed review. Your comments helped us recognize several important shortcomings in the initial draft, and the revisions benefited greatly from your guidance. We sincerely appreciate the effort you devoted to improving our work.
>
> ## 1. Response to Weakness #1 #2 #3
>
> We thank the reviewer for pointing out the missing Qwen2.5-VL-7B-Instruct results and for highlighting the broader concern regarding the diversity and generalization of our evaluation. In the initial submission, the Qwen2.5-VL-7B-Instruct results on CHAIR were indeed included, but due to suboptimal organization in the original draft, their placement created confusion. In the revised manuscript, these results are preserved and reorganized into a unified comparison table in Section 4.3, available at line 378.
>
> Following your suggestion, and consistent with the comments from other reviewers, we have substantially expanded the evaluation suite to strengthen both comprehensiveness and out-of-domain robustness:
>
> - **HallusionBench** has been added to test visual-illusion and multimodal failure cases that differ significantly from COCO-style distributions.
> - **MME** has been added to cover a wide spectrum of perception and cognition tasks.
> - **POPE (full benchmark)** now replaces the earlier hard-100 subset, ensuring alignment with prior work and eliminating potential selection bias.
> - A new **COCO-600 randomly sampled multi-label benchmark** has been introduced to broaden distributional coverage.
>
> These datasets differ markedly in visual style, task structure, and hallucination types, thereby offering a substantially broader assessment of generalization beyond the original evaluation setting. Our method remains training-free and model-agnostic, and its effectiveness across these additional benchmarks demonstrates stable inference-time behavior under distribution shift. All new results are available in Section 4.4, beginning at line 460.
>
> Collectively, these revisions address the reviewer’s concerns regarding missing results, dataset diversity, and out-of-domain generalization, ensuring a more complete and fair comparison across models and benchmarks.

---

> ### Author Response · Authors · 2025-11-25
> **Rebuttal 2: Weaknesses #4**
>
> ## 2. Response to Weakness #4
>
> We thank the reviewer for raising this valuable point. We agree that the interpretability evidence in the original submission was not presented as systematically as it could have been, and we appreciate the opportunity to clarify the scope and nature of our interpretability contribution. We would like to clarify what kind of interpretability our work aims to contribute and why we believe this contribution is appropriate.
>
> First, identifying precise functional roles for individual attention heads in large vision–language models is widely recognized as inherently challenging. Even within the mechanistic interpretability literature, both in earlier studies and in more recent probing analyses such as those by Meng et al. [5], Nanda et al. [6], and García-Carrasco et al. [7], researchers widely rely on empirical characterization. This approach focuses on behavioral evidence rather than on strict mathematical definitions or fully developed theoretical accounts. These works extract components that appear to store or manipulate specific information based on observed behavioral patterns, interventions, and quantitative signatures, not on rigid theoretical derivations.
>
> Second, in the hallucination literature, although many studies do not explicitly claim interpretability as a primary goal, their methodology implicitly relies on interpretability-oriented reasoning. For example, Yang et al.[4] identify heads that amplify language priors (“hallucination heads”), whereas others attend more faithfully to the image; SPIN[5] reduces hallucination by suppressing heads inattentive to visual evidence; VHR[6] upweights heads aligned with visual semantics. MLIH[7] uses the Attention Lens tool to summarize three empirical findings about middle-layer attention behavior. Many of these works intervene on internal components during inference. These interventions implicitly rely on a functional interpretation of those components. Such interpretations are based primarily on empirical regularities and quantitative behaviors that can be observed in practice, rather than on strict theoretical derivations or formal proofs.
>
> Against this backdrop, we believe it is reasonable to view our contributions as offering a modest but meaningful form of interpretability: we identify consistent, empirically observable patterns of functional conflict between attention heads across models and tasks, and demonstrate how these patterns can guide effective inference interventions. This is not an overclaim of theoretical interpretability, but rather an evidence-driven characterization aligned with the standard practice in both mechanistic interpretability and hallucination-mitigation work.
>
> At the same time, we fully acknowledge that our interpretability analysis remains limited in several respects. We are grateful to the reviewer for encouraging us to think more deeply about this aspect of the work. Your comment highlighted how much remains to be understood about the mechanisms underlying hallucinations, and it prompted us to consider several directions that may enrich future research.
>
> In particular, a deeper investigation of attention-head behaviors may benefit from more formal analyses of component interactions, as well as from advances in probing techniques designed specifically for LVLM architectures. We also see value in exploring broader and more systematic visualizations, together with more detailed behavioral examinations that could reveal how different kinds of hallucinations emerge from internal dynamics. These are challenging questions, but they point toward a space of inquiry that we believe is both meaningful and promising.
>
> We sincerely appreciate the reviewer’s insight, which helped us reflect on the limitations of our current presentation and consider how this line of investigation may be strengthened in future work.

---

> ### Author Response · Authors · 2025-11-25
> **Reference**
>
> [1] Meng, K., Bau, D., Andonian, A., & Belinkov, Y. (2022).
> Locating and Editing Factual Associations in GPT. NeurIPS 2023, arXiv:2202.05262.
>
> [2] Nanda, N., Chan, L., Lieberum, T., Smith, J., & Steinhardt, J. (2023).
> Progress Measures for Grokking via Mechanistic Interpretability. ICLR 2023, arXiv:2301.05217.
>
> [3] García-Carrasco, J., Maté, A., & Trujillo, J. (2024).
> Extracting Interpretable Task-Specific Circuits from Large Language Models for Faster Inference. AAAI 2025, arXiv:2412.15750.
>
> [4] Yang, Tianyun, Ziniu Li, Juan Cao, and Chang Xu.
> “Understanding and Mitigating Hallucination in Large Vision-Language Models via Modular Attribution and Intervention.” International Conference on Representation Learning (ICLR), pp. 51546–51568, 2025.
>
> [5] Sarkar, Sreetama, Yue Che, Alex Gavin, Peter A. Beerel, and Souvik Kundu.
> “Mitigating Hallucinations in Vision-Language Models through Image-Guided Head Suppression.” arXiv preprint, arXiv:2505.16411, 2025.
>
> [6] He, Jinghan, Kuan Zhu, Haiyun Guo, Junfeng Fang, Zhenglin Hua, Yuheng Jia, Ming Tang, Tat-Seng Chua, and Jinqiao Wang.
> “Cracking the Code of Hallucination in LVLMs with Vision-aware Head Divergence.” Proceedings of the 63rd Annual Meeting of the Association for Computational Linguistics (ACL), Volume 1: Long Papers, Vienna, Austria, July 2025, pp. 3488–3501.
>
> [7] Jiang, Z., Chen, J., Zhu, B., Luo, T., Shen, Y., & Yang, X. (2024).
> Devils in Middle Layers of Large Vision-Language Models: Interpreting, Detecting and Mitigating Object Hallucinations via Attention Lens. CVPR 2025, arXiv:2411.16724.

---

### Official Review · Reviewer_jmP4 · 2025-11-01

**Soundness:** 2
**Presentation:** 2
**Contribution:** 2
**Rating:** 2
**Confidence:** 4

**Summary:**

The paper targets the hallucination reduction problems in large vision-language models (LVLMs). It proposes to intervene the attention heads at a fine-grained level to mitigate LVLM hallucinations. The proposed strategy dynamically pruning or modifying the hallucination-prone attention heads, which is training-free. Experiments demonstrate improved performance in LVLM hallucination reduction.

**Strengths:**

- The paper proposes a mechanistic approach to control the VLM hallucination on the attention head level, through effective inference time intervention.

- Experimental results across three VLMs and two benchmarks show improved performance of the proposed approach.

**Weaknesses:**

- The paper writing could be improved. The novelty may also be limited given prior research in interpreting hallucination with attention heads and general mechanistic interpretability methodologies.

- Computational cost may be high. The identification of specific attention heads that are hallucination prone needs to be computed on a batch of images before inference, requiring specific head intervention.

- Different images may have different optimal head configurations. Instance level optimization on the attention head identification and intervention is time consuming, compromising the applicability of the proposed approach.

- The hyper-parameters are selected via grid search, which is also expensive considering the procedure of the approach.

- For experiments such as on CHAIR, there is no sensitivity/statistical significance analysis showing the randomness of the results, as we know the scores could vary with a large variance. The paper could benefit from testing on more recent benchmarks and models, such as InternVL, PaliGemma, Phi vision, Meta PLM, AMBER, MME, THRONE.

- The approach is image/benchmark specific, as the optimal attention heads may be different for different styles and sources of images. This impedes the practical usage of the approach.

**Questions:**

See above.

---

> ### Author Response · Authors · 2025-11-25
> **Rebuttal 1: Weaknesses #1**
>
> We sincerely thank Reviewer jmP4 for the careful reading, constructive feedback, and insightful suggestions. Your comments identified several areas where our manuscript could be strengthened, and they have substantially improved the quality, clarity, and impact of the revised version. We are grateful for the time and expertise you devoted to evaluating our work.
>
> ## 1. Response to Writing Quality and Novelty
>
> We thank the reviewer for raising concerns about writing clarity and the perceived novelty of our contribution.
>
> We agree that the initial submission did not communicate our ideas as clearly or as cohesively as it should have. In the revised manuscript, we made substantial improvements to enhance clarity and coherence. These changes primarily include a clearer presentation of Algorithm 3 (line 270+), **a complete restructuring of the experimental section** to present results in a more interpretable and logically organized manner (line 300+), and **an expanded Appendix A** that now provides detailed explanations of the experimental setup, the key engineering improvements applied to our pipeline, and a full breakdown of computational cost (line 704+).
>
> Regarding novelty, we appreciate the opportunity to clarify how our work differs from existing approaches on hallucination attribution and mechanistic interpretability. Earlier studies such as SPIN[1], VHR[2], and MLIH[3] have provided valuable insights into how attention heads may influence hallucination tendencies. These works typically characterize heads as more or less helpful in a relatively fixed way, or they apply predetermined intervention rules that do not adapt to the semantics and visual content of each specific input. As a result, these methods do not fully address the central challenge highlighted in our submission: attention heads can exhibit conflicting functional roles within the same model, and these conflicts may vary across tasks and input conditions.
>
> Our contribution differs in two key aspects, each grounded in the central problem motivating this work and not addressed in prior hallucination analyses or head-level intervention studies:
>
> First, we present empirical evidence that attention heads can exhibit systematic functional conflicts across LVLM architectures, tasks, and input distributions. Prior work has not explicitly characterized this phenomenon. Our findings highlight that a given head may contribute to faithful grounding under certain semantic–visual contexts, yet drive unreliable language decoding under other contexts. This observation motivates a more nuanced view of head-level behavior that has been largely absent in existing literature.
>
> Second, based on this observation, we develop a training-free, instance-adaptive counterbalance mechanism that adjusts the activity of both hallucination-inducing and faithful heads during inference. Unlike fixed suppression schemes or alignment-based techniques that require additional training, our method conditions the intervention on attribution signals that are tailored to each individual input. After the engineering improvements documented in Appendix A.2 and A.3, the additional inference-time cost has been reduced from an initially prohibitive level to a practically acceptable range, making the approach feasible for real-world use. Importantly, this dynamic mechanism suppresses hallucination-inducing behavior while simultaneously reinforcing faithful semantic grounding, helping mitigate hallucinations while preserving semantic informativeness, a balance that has been challenging for prior interventions.
>
> Taken together, these contributions provide a perspective on head-level behavior that is distinct from previous work and offer a practical inference-time mechanism that, to our knowledge, is not available in the existing literature. The framework connects empirical observations of internal functional conflict to a controllable and interpretable intervention strategy that remains model-agnostic, training-free, and semantically expressive across diverse evaluation.
>
> [1] Sarkar, Sreetama, Yue Che, Alex Gavin, Peter A. Beerel, and Souvik Kundu.
> “Mitigating Hallucinations in Vision-Language Models through Image-Guided Head Suppression.” arXiv preprint, arXiv:2505.16411, 2025.
>
> [2] He, Jinghan, Kuan Zhu, Haiyun Guo, Junfeng Fang, Zhenglin Hua, Yuheng Jia, Ming Tang, Tat-Seng Chua, and Jinqiao Wang.
> “Cracking the Code of Hallucination in LVLMs with Vision-aware Head Divergence.” Proceedings of the 63rd Annual Meeting of the Association for Computational Linguistics (ACL), Volume 1: Long Papers, Vienna, Austria, July 2025, pp. 3488–3501.
>
> [3] Jiang, Z., Chen, J., Zhu, B., Luo, T., Shen, Y., & Yang, X. (2024).
> Devils in Middle Layers of Large Vision-Language Models: Interpreting, Detecting and Mitigating Object Hallucinations via Attention Lens. CVPR 2025, arXiv:2411.16724.

---

> ### Author Response · Authors · 2025-11-25
> **Rebuttal 2: Weaknesses #2~#6**
>
> ## 2. Response to Computational Cost
>
> We thank the reviewer for raising the important question regarding computational cost. We acknowledge that in the initial submission, the practical overhead of our method was indeed prohibitively high. The attribution stage required close to one hour, and the hyperparameter search was performed over an inefficiently large space. Under those settings, our pipeline was difficult to deploy in real applications. Following the initial submission, we conducted extensive technical and engineering optimizations that reduced the total cost by one to two orders of magnitude.
>
> To avoid ambiguity, we reorganized the experimental section in the revised manuscript. We also rewrote the description of Algorithm 3 to clarify its role in the pipeline. In addition, we added two dedicated appendices to document the computational behavior of the method in detail:
>
> Appendix A.2 (line 786+) reports two key engineering improvements. The first improvement is **cached semantic reconstruction**, which applies to Algorithm 1 and removes hundreds of redundant forward passes. This reduces the attribution runtime from approximately one hour to around five seconds for captioning and around 0.5 seconds for VQA. The second improvement is **stabilized Bayesian initialization**, which applies to Algorithm 3 and prevents unnecessary evaluations during the early phase of optimization. This keeps the two-dimensional Bayesian search compact and efficient.
>
> Appendix A.3 (line 895+) provides a complete breakdown of the runtime for Algorithms 1 and 3. The same information can also be found in our response to reviewer icyf in the section titled “## 1. Regarding Computational Cost (Weakness #1)”.
>
> We hope this clarification resolves the concern that the original overhead limited the practical usability of our method. After the reductions achieved through the two key engineering improvements applied to the attribution stage and the optimization process, the pipeline now functions as a practical inference-time mechanism while remaining entirely training-free and model-agnostic.
>
> ## 3. Response to Evaluation
>
> We thank the reviewer for highlighting the need for a more comprehensive and diverse evaluation protocol. Following this suggestion, and consistent with feedback from other reviewers, we have substantially expanded the evaluation suite in the revised manuscript.
>
> First, we introduced a new **COCO-600 randomly sampled multi-label subset** to broaden distributional coverage within the COCO setting. This subset exposes the model to more diverse visual styles and object compositions while maintaining the standard COCO annotation structure.
> Second, we replaced the earlier hard-100 setting with the **full POPE benchmark**. This eliminates potential selection bias and aligns our evaluation protocol with established practice in the community.
> Third, we added **HallusionBench**, which focuses on visual-illusion cases and multimodal failure modes that differ substantially from COCO-style captioning and POPE’s binary object-existence format. This benchmark provides a complementary perspective on hallucination robustness.
> Finally, we incorporated **MME**, which tests a wide spectrum of perception and cognition abilities that go beyond CHAIR- and POPE-style evaluations. This inclusion allows us to examine whether the proposed approach maintains stable behavior across heterogeneous multimodal tasks.
>
> These additions collectively provide a substantially broader assessment of generalization across different visual styles, task formats, and hallucination types. All corresponding results can be found in Section 4.4 (line 460+). This expanded evaluation directly addresses the reviewer’s concern regarding dataset diversity and the need for more recent and varied benchmarks.

---

### Author Response · Authors · 2025-11-28
**Global Response to Reviewers**

We sincerely thank all reviewers for their thoughtful reading, constructive feedback, and detailed suggestions. Your comments helped us identify several areas where the exposition and organization of the manuscript could be substantially improved, and the revisions we made were directly guided by your insights. Across the four reviews, several common themes emerged—primarily concerning the clarity of the experimental section, the presentation of Algorithm 3, the computational cost of our pipeline, and notation consistency.

In response, we have made substantial revisions to strengthen clarity, structural coherence, and methodological transparency, while keeping the core technical contributions unchanged. We deeply appreciate the reviewers’ efforts and believe that the manuscript has been significantly improved thanks to your guidance.

## 1. Comprehensive restructuring of the experimental section

We reorganized Section 4 to more clearly highlight the progression from empirical findings to practical evaluation. In particular:

- The subsection _“Hallucination-Focused Attribution: An Empirical Exploration”_ is now distilled into **two clear empirical findings**, each directly supporting our head-level attribution arguments.
- We also integrated **a broader and more systematic evaluation suite**, consolidating **COCO multi-label**, **POPE**, **HallusionBench**, and **MME** under a unified narrative that highlights reliability and robustness across task forms and distributions.

This restructuring makes our contributions easier to follow and clarifies the progression from mechanistic findings to practical performance improvements.

## 2. Clear reporting of computational cost

Several reviewers requested concrete runtime measurements and a clearer discussion of computational feasibility. We now provide a unified and transparent report of all runtime-related details in Appendix A.3, including:

- runtime tables for all methods on CHAIR and POPE;
- per-instance attribution cost(≈5 s on CHAIR, ≈0.5 s on POPE);
- a description of the attribution and pruning overhead under the evaluation settings.

These additions directly address the reviewers’ concerns regarding practical feasibility.

## 3. Clearer presentation and practical refinement of Algorithm 3

Several reviewers noted that the role of Algorithm 3 and its connection to the overall pipeline were not sufficiently clear in the original submission. To address this, we revised Algorithm 3 in three specific ways:

- First, Algorithm 3 now explicitly points to the steps where Algorithm 1 (attribution) and Algorithm 2 (dynamic pruning) are invoked.
  This structural clarification makes the flow of information and the dependence between the components much easier to follow.

- Second, we document in Appendix A.2 two key engineering improvements that make Algorithm 3 substantially faster and more stable in practice. These implementation refinements reduce inference-time overhead and provide a clearer picture of how the pipeline behaves in real-world deployment.

- Third, in Section 4 we clarify the scope and usage of Algorithm 3 in all experiments, ensuring that the meaning of “Ours” is unambiguous.
  This resolves the confusion raised by multiple reviewers regarding when Algorithm 3 is applied and how it differs from the generalization experiments.

## 4. Notation and textual clarifications

We made several small but important adjustments to improve clarity and avoid potential ambiguity in the revised manuscript:

- (a) The cumulative-energy threshold in Eq. (3) is now denoted by **τ**, ensuring that **α** is used only as the coefficient of the SpectralVar term and eliminating the earlier symbol conflict.
- (b) Several expressions involving the spectral regularizers were rewritten in a clearer and more explicit form. These changes affect only the presentation of the formulas, not their mathematical meaning or implementation.
- (c) A sentence in the Motivation was revised to avoid overstating the causal role of attention-head conflicts, making the phrasing more precise.
- (d) We also made minor organizational adjustments in the Related Work section.
  A few methods previously listed under external modules are now placed together with other decoding-guided approaches, and the newly added multi-agent post-hoc method is introduced alongside existing expert-based systems. These edits improve narrative consistency and make the distinctions between different families of approaches clearer to the reader.

We have provided detailed, point-by-point responses to all concerns raised by the reviewers, and we hope the revisions clearly address each issue. Should any additional questions arise, we will actively respond throughout the remaining discussion period to ensure that all points are fully clarified. We sincerely hope the revised manuscript reflects our contributions more clearly, and we truly appreciate the reviewers’ guidance throughout this process.

---

> ### Author Response · Authors · 2025-11-28
> **Full Changelog for Rebuttal Submission**
>
> - **Reorganized Section 4: Experiment (lines 300–525)** to refine the flow from empirical findings to evaluation results.
> - **Updated Section 4.1 (Experimental Setup)** by clarifying the usage and scope of “Ours” across all evaluations (line 304). Additional clarifications were added throughout the experimental section wherever ambiguity could arise.
> - **Condensed the extended analysis in Section 4.3 into two empirical findings**, and consolidated the hallucination-focused comparison with SOTA methods into a single table (Table 3, line 378) to more clearly support the analysis.
> - **Added a new Section 4.4: Broad Evaluation Across Diverse Hallucination Benchmarks (line 460)**, covering COCO multi-label, full POPE, HallusionBench, and MME under a unified evaluation structure. This section also replaced the POPE hard-100 subset with the full POPE benchmark. All corresponding experimental tables were added or updated.
> - **Reorganized the appendix**, introducing a new Appendix A (line 704+), which includes:
>   (A.1) detailed experimental setups for all methods;
>   (A.2) two engineering improvements to Algorithm 3;
>   (A.3) a full runtime breakdown with corresponding tables.
> - **Revised Algorithm 3** to explicitly reference the steps where Algorithm 1 (attribution) and Algorithm 2 (dynamic pruning) are invoked (line 270).
> - **Updated Eq. (3)** by replacing the cumulative-energy threshold symbol **α** with **τ** (line 201).
> - **Rewrote expressions describing InfoSpectralScore** for clarity without altering their mathematical meaning (line 239).
> - **Revised a sentence in the Motivation** to avoid overstating the causal role of attention-head conflicts (line 124).
> - **Refined the categorization of related work** to distinguish decoding-guided approaches from external expert–based methods, and added a citation to a post-hoc multi-agent correction method to strengthen the coverage of the latter category (lines 100–107).
> - **Corrected the SPIN results on Shikra in Table 3 (line 392)** and provided detailed parameter explanations for this correction in Appendix A.1 (line 718).

---

### Author Response · Authors · 2025-12-02
**Summary to Area Chair**

Dear Area Chair,
Thank you for your time and for considering our submission. Following the program chairs’ guidance, we provide a brief summary of the key clarifications and additions in our rebuttal and revised manuscript. Our responses directly address the core concerns raised by the reviewers:

## Core contributions and motivation

Our work addresses the challenge that instance-level hallucinations can arise from conflicting behaviors across attention heads.
We contribute **(i) InfoSpectralScore**, a semantics-grounded head-level attribution metric that reliably identifies both hallucination-prone and faithful heads, and **(ii) a dynamic, training-free counteractive pruning mechanism** that adaptively suppresses the former while reinforcing the latter during inference. Together, these components enable fine-grained, per-instance control of LVLMs without retraining. Extensive experiments further confirm that **our approach reduces hallucination while preserving semantic expressivity**, and that these effects remain robust across models, data distributions, and evaluation tasks.

## Computational cost and practicality

Since computational overhead was a common concern in the reviews, we made targeted engineering changes to reduce per-instance cost. We streamlined the overall pipeline with lightweight cached attribution and a compact BO search, keeping the end-to-end runtime well within a feasible range for real deployments. In particular, cached attribution reduces the attribution stage to only a few seconds per instance.

## Strengthened empirical support (Sections 4.3 & 4.4)

In Section 4.3, we reorganized our analysis into **two empirical findings** that reinforce our head-level attribution arguments, showing that attribution derived from hallucination-focused cases provides more informative and stable signals and that leveraging these signals enables more robust pruning behavior across models and distribution shifts. We also clarified where Algorithm 3 is applied and kept the definition of “Ours” consistent across sections for direct comparability. In Section 4.4, following reviewers’ requests, we added broad evaluations on a **COCO multi-label random subset**, **the full POPE benchmark**, **HallusionBench**, and **MME**, demonstrating consistent improvements across caption-level, object-level, illusion-focused, and perception-style benchmarks.

Due to an unexpected interruption of the discussion period, reviewers had no opportunity to respond further to our clarifications and new results. Even so, we believe that our rebuttal and revision directly address the reviewers’ main concerns and substantially strengthen the paper. We respectfully ask that you consider the updated manuscript and clarifications, and we trust your careful judgment in making the final recommendation.

**Thank you very much for your time and consideration.**

---

### Meta-Review · Area_Chair_rNDX · 2026-01-02

**Summary:**

The reviewers raised concerns about the strength of the experimental evidence and the substantiation of key claims. The added experiments only partially address these issues and do not fully support the conclusions drawn. In addition, the interpretability claims are weakly supported, relying largely on empirical intuition.

**Reviewer Concerns:**

Major concerns still remain unresolved. The interpretability claims are still weak, as the evidence relies mainly on empirical intuition rather than systematic analysis. In addition, the conclusions drawn from Table 4 remain overstated, particularly regarding recall and overall advantage, and the comparison with stronger baselines is insufficient.

**Reviewer Scores:**

I believe the reviewers’ scores would largely remain unchanged.  Reviewers who were initially critical would likely maintain their original assessments, at most shifting to a borderline reject rather than a clear accept.

---

### Decision · Program_Chairs · 2026-01-26

Reject